# The Stability-Efficiency Dilemma: Investigating Sequence Length Warmup for Training GPT Models

**Conglong Li**
Microsoft
conglong.li@microsoft.com

**Minjia Zhang**
Microsoft
minjiaz@microsoft.com

**Yuxiong He**
Microsoft
yuxhe@microsoft.com

## Abstract

Recent works have demonstrated great success in pre-training large-scale autoregressive language models (e.g., GPT-3) on massive GPUs. To reduce the wall-clock training time, a common practice is to increase the batch size and learning rate. However, such practice is often brittle and leads to a so-called stability-efficiency dilemma: increasing the batch sizes and learning rates leads to better training efficiency but can also result in training instability, leading to poor generalization accuracy or failed runs. To better understand this phenomenon, we conduct an in-depth analysis on large-scale pre-training experiments replicating the GPT-2 model with public dataset. We find that there is a strong correlation between training instability and extreme values of gradient variance. We further identify that samples with long sequence lengths contribute to these extreme gradient variance values, especially at the beginning of the training, indicating that long sequence length can be a main source of training instability.

Based on the analysis, we present a simple yet effective Sequence Length Warmup method that aims to solve the training stability-efficiency dilemma by avoiding extreme gradient variance values. Moreover, we present a lightweight tuning strategy that allows us to tune our method with just a small portion of the expensive full training. Experiments replicating GPT-2 models (117M and 1.5B) show that our approach enables stable training with 8x larger batch size and 4x larger learning rate, whereas the baseline approach struggles with training instability. To achieve the same or better zero-shot evaluation results, our method reduces the required number of training tokens and wall clock time by up to 2.2x and 3.7x, respectively. Experiments replicating GPT-3 model (125M) show that our approach enables stable training with 8x larger batch size and 40x larger learning rate, and retains 99% of the zero-shot accuracy on 11 tasks using 10x less data and 17x less time compared to the original GPT-3 training recipe, while the baseline diverges under the same settings and only retain 95% of accuracy under lower learning rate.

## 1 Introduction

Large-scale Transformer-based language models have achieved great success in many natural language processing tasks [46, 11]. Among them, large-scale autoregressive models, such as GPT-3 [6], have attracted lots of attention due to their superior performance on zero-shot generalization, i.e., they can perform a wide range of tasks that they are not explicitly trained on. However, pre-training GPT models raises huge challenges on training efficiency and less-discussed training instability issues. On the efficiency side, as the model size continues to grow from a few hundreds of millions (e.g., GPT [31]), to billion-scale parameters (1.5B GPT-2 [33]), and to more recent hundreds of billions of parameters (175B GPT-3 [6]), the training cost also increases exponentially: it requires 9.2 days

---

This paper was previously titled "Curriculum Learning: A Regularization Method for Efficient and Stable Billion-Scale GPT Model Pre-Training" in an early arxiv preprint version[22].

36th Conference on Neural Information Processing Systems (NeurIPS 2022).

on 512 V100 GPUs to train a 8.3B GPT-2 [40], and 47.8 days on 2240 A100 GPUs to train a 530B GPT-3-style model [26]. Such a long training time makes it almost infeasible for most research labs and practitioners to reproduce these models. Various solutions have been proposed to reduce the training wall clock time of these large models [40, 25, 35]. However, many solutions require using more GPUs or sophisticated system techniques.

In this work, we investigate speeding up the pre-training of GPT-style models via exploiting data efficiency, not at the cost of excessive hardware resources. In particular, in a distributed training environment, increasing the batch sizes and/or using more aggressive learning rates can make the model converge faster [41]. However, it has been observed that larger batch sizes and learning rates can make large-scale GPT model training more difficult, e.g., causing training instability that leads to divergence or slow convergence [49, 9]. To investigate this training instability issue, we conduct a thorough study of the GPT-2 pre-training task [33, 40] with different models sizes under various batch sizes, learning rates, and sequence lengths. We find a **stability-efficiency dilemma**:

- A larger batch size (and larger learning rate) increases the per-iteration computational efficiency but with increasing likelihood of training instability and even divergence.
- A smaller batch size makes the training more stable but decreases the per-step computation efficiency significantly.

We find it difficult to overcome this dilemma by existing techniques such as extra gradient clipping. More recent proposed techniques such as batch size warmup proposed in [6] does not provide stability benefit in our evaluations. Recently, Shortformer[30] shows that by adding an additional first training stage with a shorter sequence length, language models can achieve the same dev. set perplexity with shorter total training time. However, (1) its main focus is to purely improve training efficiency instead of solving the efficiency-stability dilemma, and our evaluations show that Shortformer's 2-stage approach is insufficient for overcoming the training instability issue for large models (Section 5.1), (2) it is primarily evaluated on small scale transformer models (247M) on WikiText datasets (103M tokens) without considering large-scale generative models like GPT with hundreds or even billions of parameters, and (3) it does not discuss how to choose some of the important hyperparameters, which is very expensive to figure out for large-scale model training.

Inspired by Shortformer, we investigate the importance of sequence length in training GPT models and find that it plays an important role in both training stability and efficiency. Based on our investigation, we propose a simple yet effective method called **Sequence Length Warmup (SLW)**, which starts training with short sequences and gradually increases the length. We observe that our approach enables stable and efficient training with much larger batch sizes and learning rates than baseline approaches. Specifically, we make the following contributions: (1) We conduct an extensive study of the GPT-2 pre-training task, which provides detailed insights about the training stability-efficiency dilemma, the correlation between instability and gradient variance outliers, and how sequence length plays a critical role (Section 3). (2) Based on the study, we present a simple yet effective sequence length warmup method for GPT-style model (and autoregressive model in general) that enables stable training with improved training efficiency. We also identify a lightweight hyperparameter tuning strategy for the approach, which identifies promising hyperparameters by only incurring a small fraction of the expensive total pre-training cost (Section 4). The implementation of our approach as well as the necessary changes to the GPT-2/3 pre-training framework has been open sourced in a deep learning optimization library called DeepSpeed[1]. (3) We conduct large-scale experiments to demonstrate the proposed work's ability to provide superior training stability and efficiency at the same time (Section 5). Our empirical results show that:

- SLW enables stable and efficient training with 8x larger batch size and 4x larger learning rate on GPT-2 (117M and 1.5B) models with public datasets, while the baseline and related works struggle with instability under the same settings. To achieve the same or better zero-shot WikiText-103/LAMBADA evaluation results at the end of training, SLW reduces the required number of training tokens and wall clock time by up to 2.2x and 3.7x, respectively.
- On GPT-3 model (125M) pre-training we study an even more aggressive training scenario where only 10% of data can be used. Our method, with 8x larger batch size and 40x larger learning rate than the original GPT-3 training recipe, is able to maintain the training stability, retaining 99% of the zero-shot accuracy on 11 evaluation tasks, and use 10x less data and 17x less time. Without our

---

[1]https://github.com/microsoft/DeepSpeed, https://www.deepspeed.ai/

method, the baseline has unrecoverable divergence under the same settings, and can only retain 95% of the zero-shot accuracy after lowering learning rate to 30x.

## 2 Background and Related Work

**Language Model Pre-training:** The accuracy of transformer-based language models grows substantially with its model size [31, 33, 6]. Today, a large language model such as GPT-3 [6] contains up to 175B parameters, and recent studies show that model accuracy can continue to improve with even larger model sizes [16]. However, training these large models often incurs excessively long training time and training difficulties [6]. Therefore, there are a lot of demands of performing efficient and stable training for these models. To have the pre-training finished in a reasonable amount of time, the most common way is to leverage data parallelism to train models on multiple GPUs. However, the speedup gains often saturate beyond a few tens of GPUs, because communication becomes the major bottleneck, i.e., the workers will spend more time communicating gradients than computing them, as the number of GPUs increases. To mitigate this bottleneck, recent works such as 1-bit Adam [43] have studied gradient compression and demonstrate their effectiveness against auto-encoding models such as BERT [11]. An alternative approach to alleviate these overheads is to use large batch sizes. For example, LAMB [52] and 1-bit LAMB [21] enable stable and efficient distributed BERT pre-training with batch size up to 64K/32K (for sequence length 128/512, i.e., 8M/16M tokens per batch) while maintaining the sample-wise convergence speed. For encoder-decoder models, T5 [34] uses batch size up to 2K (for sequence length 512, i.e., 1M tokens per batch). For autoregressive models such as the GPT family [31, 33, 6], existing works use batch size up to 1.6K (for sequence length 2K, i.e, 3.2M tokens per batch). Despite the benefit of reduced communication overhead, large-batch training is sensitive to hyperparameters and often leads to issues such as slow convergence, training instabilities, and model divergence. For example, recently a research project shared that they are dealing with challenging training instability issues when pre-training a 104B GPT-style model with batch size 2K [49], and another work on a 540B model with batch size 2K observed spikes in the loss roughly 20 times during training, despite the fact that gradient clipping was enabled [9].

**Curriculum Learning:** Our method can be viewed as a kind of curriculum learning (CL) [12, 39, 2], which presents easier/simpler examples earlier during training and gradually increases the sample difficulties[2]. Comparing with traditional CL works which focus on solely improving the convergence speed under the same batch size, learning rate and other hyperparameters, our work is motivated by the stability-efficiency dilemma and we aim to achieve both efficient convergence and better stability by enabling stable training with more aggressive hyperparameters. To our knowledge, we are the first to investigate and confirm that certain curriculum learning method can provide a dual stability-efficiency benefit.

In the NLP area, most of the curriculum learning works focus on small-scale one-stage tasks and downstream fine-tuning tasks, such as neural machine translation (NMT) [18, 5, 56, 29, 57] and natural language understanding (NLU) [36, 37, 44, 50]. There are also a few works explore curriculum learning for language model pre-training [30, 55, 7]. These works show that curriculum learning can improve convergence speed, reduce training time, and improve accuracy under the same training hyperparameters as baseline. In these works, the curriculum difficulty metrics for each training sample are usually defined as the sentence length, vocabulary frequency, the inference loss on smaller/cheaper models, or based on self-paced learning [19]. For the pacing function (i.e., to decide the curriculum difficulty range when sampling next training data batch), these works usually use fixed predefined functions (e.g., gradually increase difficulty upper bound by linear, root, and exponential functions), bucketing heuristics (group data with similar difficulties, and sample from a subset of buckets every time), or based on self-paced learning.

## 3 GPT-2 Pre-training Stability-Efficiency Analysis

In this section we perform an in-depth analysis of pre-training tasks (without our method) replicating the GPT-2 models with public data. We follow the training pipeline from the NVIDIA Megatron-LM work [40][3]. All of the experiments are performed on 128 NVIDIA V100 GPUs (32GB memory). There are 16 nodes and 8 GPUs per node. GPUs inside the same node are connected by NVLink 2.0, and nodes are connected by a 100 Gigabit InfiniBand EDR inter-node network. We evaluate

---

[2]The shorter sequences are not necessarily easier but can be viewed as simpler examples since there are less context to embed.

[3]https://github.com/NVIDIA/Megatron-LM

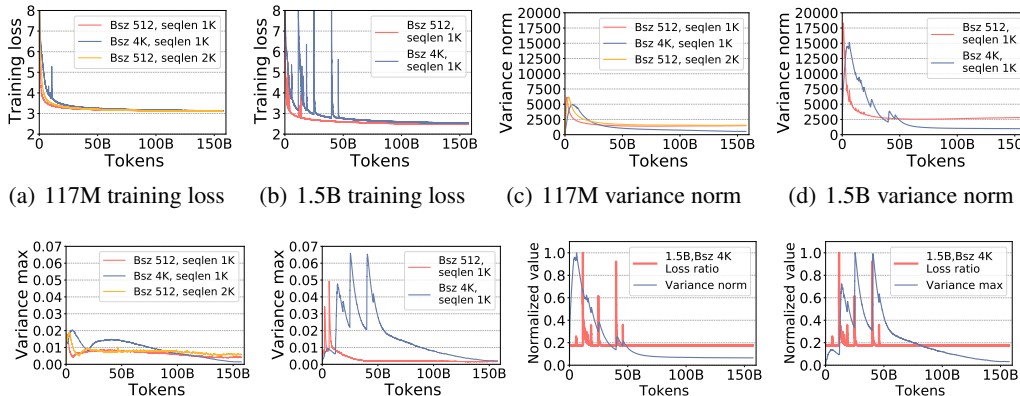

| (a) 117M training loss | (b) 1.5B training loss | (c) 117M variance norm | (d) 1.5B variance norm |
|---|---|---|---|
| (e) 117M variance max | (f) 1.5B variance max | (g) Var norm correlation | (h) Var max correlation |

Figure 1: Training loss, Adam variance norm/max element, and correlations between loss spikes and variance norm/max during GPT-2 pre-training (without the proposed method) under different model sizes, batch sizes (and LR), and sequence lengths. In Appendix A.1 we plot the same figure but zoom in the first 30B tokens.

two GPT-2 model sizes from the original GPT-2 work [33]: 117M parameters (12 layers, 768 hidden size, 12 attention heads) and 1.5B parameters (48 layers, 1600 hidden size, 25 attention heads). For training data, we collect and use the same dataset blend as the Megatron-LM work: Wikipedia [11], CC-Stories [45], RealNews [54], and OpenWebtext [32].

We evaluate two sets of training parameters. The first set follows the Megatron-LM work: batch size 512, 300K total training steps (157B tokens), and learning rate $1.5 \times 10^{-4}$ with a linear warmup of 3K steps and a single cycle cosine decay over the remaining 297K steps ($1 \times 10^{-5}$ min. learning rate). The second parameter set tests a more aggressive training strategy: batch size 4K ($8\times$ larger), 37.5K total training steps (157B tokens[4]), and learning rate $6 \times 10^{-4}$ ($4\times$ larger) with a linear warmup of 3K steps and a single cycle cosine decay over the remaining 34.5K steps (same min. learning rate). For sequence length/context size, we mainly use 1K which is the default for GPT-2. But we also test 2K (on the 117M model with batch size

Table 1: Measuring training instability by the ratio between the current step training loss and the minimum loss among all previous steps. Larger ratios (esp. those greatly larger than 1.0) indicate larger training instability/divergence. Proposed work (SLW) and related works (last two rows) are discussed in Section 5.

| Case | Pre-training parameters | #steps with loss ratio > 1.2 (% of total steps) | max loss ratio |
|---|---|---|---|
| **117M:** | | | |
| 1: Baseline | bsz512-seqlen1K | 0 (0.0%) | 1.05 |
| 2: SLW 60K | bsz512-seqlen1K | 0 (0.0%) | 1.06 |
| 3: Baseline | bsz4K-seqlen1K | 22 (0.06%) | 1.42 |
| 4: SLW 20K | bsz4K-seqlen1K | 0 (0.0%) | 1.02 |
| 5: Baseline | bsz512-seqlen2K | 0 (0.0%) | 1.04 |
| 6: SLW 110K | bsz512-seqlen2K | 0 (0.0%) | 1.04 |
| **1.5B:** | | | |
| 7: Baseline | bsz512-seqlen1K | 114 (0.04%) | 2.10 |
| 8: SLW 270K | bsz512-seqlen1K | 0 (0.0%) | 1.06 |
| 9: Baseline | bsz4K-seqlen1K | 1381 (3.68%) | 5.65 |
| 10: SLW 45K | bsz4K-seqlen1K | 0 (0.0%) | 1.02 |
| 11: Shortformer | bsz4K-seqlen1K | 219 (0.4%) | 2.86 |
| 12: Bsz Warmup | bsz4K-seqlen1K | 1179 (2.01%) | 4.32 |

512 and 157B tokens) which is the default for GPT-3. All experiments are performed with mixed precision/FP16 training, Adam optimizer ($\beta_1 = 0.9$, $\beta_2 = 0.999$, $\epsilon = 1 \times 10^{-8}$) [17], 0.01 weight decay, same random seed, and gradient clipping at 1.0. For both batch sizes we use the same number of gpus (128). It is true that under fewer nodes, smaller batch sizes can also achieve good computation efficiency. However, in practice, the goal of a training task is usually "given a fixed number of hardwares, how to train the model in the fastest wall clock time". And given the increasing model sizes, pre-training on hundreds of GPUs is not uncommon. Thus we believe that using the same hardware resources is a fair comparison.

**The stability-efficiency dilemma:** Figure 1(a) and 1(b) present the training loss curves of 5 baseline cases under different model sizes, batch sizes (and LR), and sequence lengths. At 117M, the baseline has a few training loss spikes at batch size 4K. At 1.5B, the baseline has many loss spikes when training with either batch size 512 or 4K. As an indicative measurement to quantitatively study training instability, we define "loss ratio" which measures the ratio between the current step training loss and the minimum loss among all previous steps. A ratio larger than 1.0 means that current step's loss is larger than the previous minimum loss, thus larger ratio indicates a larger loss spike and training instability. Table 1 summarizes the number of steps with loss ratio larger than 1.2, and the

---

[4]For pre-training it is common to keep the number of training tokens the same for fair comparison.

Table 2: Zero-shot evaluation of the trained models on the WikiText-103 and LAMBADA datasets, following the evaluation methodology from [40]. Case 2 to 9 are compared with case 1, and case 11 to 17 are compared with case 10. Proposed work (SLW) and related works (16, 17) are discussed in Section 5.

| | Case | Pre-training parameters | Training steps | Training tokens | Training time | WikiText PPL ↓ | LAMBADA accuracy ↑ |
|---|---|---|---|---|---|---|---|
| **117M:** | 1: Baseline | bsz512-seqlen1K | 300K | 157B | 37Hr | 27.78 | 33.19% |
| | 2: SLW 60K | bsz512-seqlen1K | 200K | 89B (1.8x) | 20Hr (1.9x) | **27.74** | **34.78%** |
| | 3: SLW 60K | bsz512-seqlen1K | 330K | 157B (1x) | 33Hr (1.1x) | **27.01** | **34.41%** |
| | 4: Baseline | bsz4K-seqlen1K | 37.5K | 157B (1x) | 16Hr (2.3x) | 28.09 | 32.54% |
| | 5: SLW 30K | bsz4K-seqlen1K | 37K | 92B (1.7x) | 10Hr (**3.7x**) | **27.77** | **33.40%** |
| | 6: SLW 30K | bsz4K-seqlen1K | 52.5K | 157B (1x) | 16Hr (2.3x) | **27.15** | **34.16%** |
| | 7: Baseline | bsz512-seqlen2K | 150K | 157B (1x) | 32Hr (1.2x) | 28.19 | 32.99% |
| | 8: SLW 110K | bsz512-seqlen2K | 122.5K | 71B (**2.2x**) | 15Hr (2.5x) | **27.06** | **33.24%** |
| | 9: SLW 110K | bsz512-seqlen2K | 205K | 157B (1x) | 31Hr (1.2x) | **26.03** | **34.58%** |
| **1.5B:** | 10: Baseline | bsz512-seqlen1K | 300K | 157B | 341Hr | 13.89 | 57.29% |
| | 11: SLW 270K | bsz512-seqlen1K | 360K | 122B (1.3x) | 286Hr (1.2x) | **13.89** | **57.38%** |
| | 12: SLW 270K | bsz512-seqlen1K | 428K | 157B (1x) | 364Hr (0.9x) | **13.88** | **57.89%** |
| | 13: Baseline | bsz4K-seqlen1K | 37.5K | 157B (1x) | 151Hr (2.3x) | 14.76 | 55.06% |
| | 14: SLW 45K | bsz4K-seqlen1K | 50K | 121B (1.3x) | 121Hr (2.8x) | **13.88** | **58.20%** |
| | 15: SLW 45K | bsz4K-seqlen1K | 58.8K | 157B (1x) | 155Hr (2.2x) | **13.72** | **58.47%** |
| | 16: Shortformer | bsz4K-seqlen1K | 55K | 157B (1x) | 162Hr (2.1x) | 14.14 | 57.23% |
| | 17: Bsz Warmup | bsz4K-seqlen1K | 58.8K | 157B (1x) | 165Hr (2.1x) | 14.21 | 56.36% |
| **Reference works:** | 18: Original GPT-2 117M [33], different data | | | | | 37.50 | 45.99% |
| | 19: Original GPT-2 1.5B [33], different data | | | | | 17.48 | 63.24% |
| | 20: Megatron-LM GPT-2 355M [33], same data | | | | | 19.31 | 45.18% |
| | 21: Megatron-LM GPT-2 2.5B [33], same data | | | | | 12.76 | 61.73% |

maximum loss ratio during the training. At 117M model size only the baseline with batch size 4K has high loss ratios up to 1.421. At 1.5B model size the baseline with both batch size 512 and 4K has much more steps with large loss ratios, and with the maximum loss ratio as high as 5.65. Baseline with batch size 4K is less stable than baseline with batch size 512, indicating that larger batch sizes (combined with larger learning rates) could lead to more training instability risks. In Appendix A.3.1 we show that larger learning rates under the same batch size could also increase training instability.

Training instability are undesirable because (1) it could lead to divergence that never recover as in [49] and our GPT-3 experiments (Section 5.2); (2) in our GPT-2 case it leads to worse convergence, validation loss, and zero-shot downstream task accuracy. Table 2 summarizes the zero-shot WikiText-103/LAMBADA evaluation results. For both 117M and 1.5B models, increasing baseline's batch size (and LR) or sequence length leads to training instabilities and loss spikes, and it requires a nontrivial number of training steps/tokens to recover the training loss back to a normal level (e.g., Figure 1(b)). These training "detours" slow down the learning and eventually lead to worse evaluation results (e.g., case 13 vs case 10 in Table 2). On the other hand, increasing batch size (and LR) or sequence length improves training efficiency, reducing the training time by up to 2.3x under the same number of training tokens (case 1, 4, 10, 13).

Overall, the above observations demonstrate the stability-efficiency dilemma for baseline pre-training: the training is more stable and can achieve better final generalization, but presumably with poorer training efficiency under smaller batch size/learning rate/sequence length; increasing them leads to better training efficiency, but with lower stability and worse generalization.

**The correlation between instability and gradient variance outliers:** For stochastic gradient optimization, when the gradient variance is large, the algorithm might spend much time bouncing around, leading to slower convergence and potential divergence [48]. Previous studies show that variance reduction methods improve training stability in areas such as reinforcement learning [24, 8, 1]. Figure 1(c), 1(d), 1(e) and 1(f) plot the $l_1$ norm and max element of Adam's variance state ($\sqrt{v_t}$, where $v_t = \beta_2 v_{t-1} + (1 - \beta_2)(g_t)^2$)[5]. When baseline's batch size increases, the max variance norm decreases but the max element increases. Comparing GPT-2 117M and 1.5B cases, larger model size leads to larger variance norm and max element. When sequence length increases for the GPT-2 117M case, the variance norm stays the same but the max element increases.

To further study the link between instability and gradient variance, Figure 1(g) and 1(h) plot the loss ratio (defined earlier in this section) and gradient variance norm/max element (all normalized by max value) for the most unstable 1.5B baseline with 4K batch size. Results show that when training loss spike happens and loss ratio increases, the gradient variance norm/max also increase (especially the max outliers). Table 3 presents the Pearson correlation coefficient calculations, which demonstrate

---

[5]We use $l_1$ norm to avoid outlier amplification.

Table 3: Pearson correlation coefficient (with range (-1, 1)) between loss ratio and gradient variance norm/max. Low p-value indicates that the correlation is statistically significant.

| | Pearson correlation coefficient | p-value |
|---|---|---|
| Loss ratio vs Gradient variance norm | 0.23 | 0.0 |
| Loss ratio vs Gradient variance max | 0.26 | 0.0 |

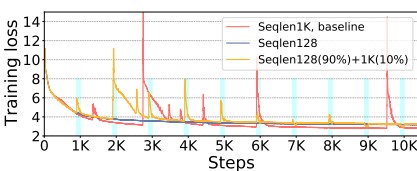

Figure 2: Step-wise training loss during GPT-2 1.5B pre-training (first 10K steps only) with batch size 4K, comparing seqlen 1K (baseline), seqlen 128, and mixed seqlen of 128+1K (1K seqlen used at the cyan areas).

a statistically significant positive correlation between loss ratio and gradient variance norm/max. Overall, our analysis shows that training instability has a strong correlation with gradient variance norm and (especially) max element outliers.

**Length of early data sequences is critical to training stability:** Aiming to solve the stability-efficiency dilemma we first tried traditional methods such as increasing gradient clipping, but it does not fully resolve the instability issue (Appendix A.3.2). Seeing that in Figure 1 the training instability mostly happens at the first half of training, we then explored whether we can solve the issue by gradually increasing any of the batch size/learning rate/sequence length during training. We already employed the same learning rate warmup mechanism used by existing GPT-2 and GPT-3 works [33, 40, 6]. We tried the batch size warmup method proposed in GPT-3 work [6], but the instability issue still appears when increasing the batch size (Section 5.1). Our investigation on the sequence length leads to interesting findings, where we find that sequence lengths play a critical role in training instability. Figure 2 presents the training loss curve of the most unstable GPT-2 1.5B pre-training with batch size 4K and seqlen 1K, together with another two artificial settings: one with seqlen 128, the other with mixed seqlen where we feed 900 steps of seqlen 128 then 100 steps of seqlen 1K in every 1K steps. The seqlen 128 case has no instability issue, even with large model size/batch size/learning rate. The mixed seqlen case has instability issues, and (1) they mostly happen when we switch to seqlen 1K (e.g., at step 900, 1900, 2900...); (2) they mostly happen during the first 5K steps, and after that it becomes more stable than the seqlen 1K case. These observations indicate that training instability is strongly correlated with early long sequence lengths, which motivates us to explore the sequence length warmup method described in the next section, and evaluations in Section 5 will demonstrate how this method provides a gradient variance reduction effect and solves the stability-efficiency dilemma in our experiments.

## 4 The Sequence Length Warmup Method

The analysis in last section about training instability and sequence lengths motivates us to explore sequence length warmup methods: the model needs to start learning with short sequence length for more stable training, then gradually increase the length when training becomes more stable so that the model can still learn from longer contextual information to achieve better final model accuracy.

The sequence length warmup strategy depends on two factors: how to support variable sequence length during training and how to adaptively decide the sequence length for each iteration (the pacing function). For the first component, we develop an efficient truncation-based implementation: For the baseline GPT-2 pre-training, the raw text inputs are indexed into sequences with the same length before training, so that the model can efficiently retrieve a batch of fixed-length sequences regardless of the actual sentence boundaries. It's possible to index the text inputs based on all possible sequence lengths, but that adds significant amount of overhead due to the massive pre-training data. To avoid the large indexing overhead, we take a lightweight approach: our implementation still lets the dataloader index the raw text into only the full sequence length. At each training step, our method uses pacing function to determine the sequence length and then truncates the full-length sequences to obtain a modified version of the mini-batch for training. It is true that this truncation-based implementation will drop some data in the current step. However, with some implementation changes, it's possible to record the index of dropped data and use them in future steps.

We define the pacing function as a step-wise linear function with the following properties: Given a starting sequence length $seqlen_s$, an ending sequence length $seqlen_e$ (full sequence length), and a total duration $T$ (number of steps), the sequence length used for the training batch at step $t$ is $seqlen_t = seqlen_s + (seqlen_e - seqlen_s) \times min(\frac{t}{T}, 1)$. Besides step-wise linear, we also explored 3 other pacing functions: i) We tried a discrete 2-stage pacing function from [30], but it leads

to unstable training and worse convergence (Section 5.1). ii) We tried a step-wise root function ($seqlen_t = seqlen_s + (seqlen_e - seqlen_s) \times min((\frac{t}{T})^r, 1)$, where $r$ is the root degree), which performs similar to linear but requires one extra hyperparameter. iii) We tried an adaptive pacing function based on training/validation losses, which also performs similar and requires extra tuning.

**Pacing function analysis and tuning strategy:** To study the impact of our approach's pacing function, we set the starting sequence length ($seqlen_s$) fixed at 8 and perform a grid search for the pacing function duration ($T$) on the GPT-2 117M case full training (details in Appendix A.4). After full trainings we perform evaluation on pretrain data test set and WikiText/LAMBADA zero-shot tasks to confirm which duration $T$ provides the best accuracy performance. All the cases have quite comparable evaluation results, indicat-

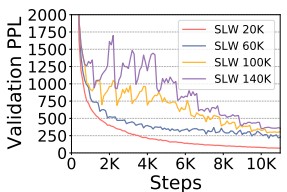
(a) Step-wise validation perplexity (beginning of training)

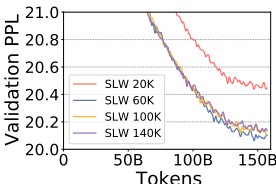
(b) Token-wise validation perplexity (end of training)

Figure 3: Validation perplexity during GPT-2 117M seqlen 1K pre-training with batch size 512 and different duration $T$. ("SLW 20K" means proposed approach with $T$=20K steps).

ing that the performance is not very sensitive to the duration $T$ within a reasonable range. This grid search sheds light on a low-cost tuning strategy: we find that (for GPT-2 117M training with batch size 512 and 4K) the best duration $T$ is the longest duration that does not have significant validation perplexity fluctuation during the first 10K steps (i.e., a few multiples of the LR warmup steps). In on our study, this "significant fluctuation" can be defined as "whether the perplexity value becomes larger than 1.3x of the previous best perplexity". In Figure 3(a) the SLW 60K is the longest duration we tested that does not have significant validation fluctuation during the first 10K steps. In Figure 3(b) and Appendix A.4 SLW 60K does provide the best final validation perplexity, best final test perplexity, and second best eval results. Since it does not require training the model until full convergence, this heuristic greatly reduces the hyperparameter tuning cost of our approach. Another grid search on the starting sequence length $seqlen_s$ shows that it's generally better to set it as small as possible, to maximize the stability and convergence speedup benefit. However, if the validation perplexity has significant fluctuation near the starting sequence length, increasing $seqlen_s$ would lead to better convergence.

Overall, the low-cost tuning strategy can be summarized as: (1) Start with $seqlen_s = 8$ and $T = $ a few multiples of LR warmup steps. (2) Increase $seqlen_s$ until the validation perplexity no longer has significant fluctuation at the very beginning. (3) Perform a binary search to find the largest $T$ that does not have significant validation perplexity fluctuation during the first few multiples of LR warmup steps. This tuning strategy relies only on the validation set and does not require test set or downstream task evaluation. For the GPT-2 1.5B and GPT-3 125M models, we used this strategy to tune $T$ and $seqlen_s$ for the pacing function, and results show that this low-cost tuning strategy could provide similar stability-efficiency benefit as grid search on full training runs (GPT-2 117M case).

## 5 Evaluation

### 5.1 GPT-2 experiments

For GPT-2 model, dataset, and hardware, we follow the same methodology in Section 3. For proposed work's pacing function configurations (Section 4), we use $seqlen_s = 8/64$ (for 117M/1.5B model based on tuning) and $seqlen_e = 1K/2K$ (full sequence length). To fully utilize the NVIDIA Tensor Core acceleration, we add a $seqlen_t = seqlen_t - (seqlen_t \bmod 8)$ postprocessing to make sure the sequence length is always a multiple of 8. For the total duration $T$, we tune this parameter (grid search for 117M and low-cost tuning for 1.5B) for each case. For the training parameters, for our approach we use the same shared parameters as the baseline except two parameters: 1) Because during sequence length warmup the number of tokens in a data batch is smaller, we modify the training termination condition so that all cases stop when reaching the same 157B training tokens. 2) Because of 1), proposed approach now has more training steps, which make it necessary to modify the learning rate decay schedule to have a fair comparison with the baseline. We change the learning rate decay to token-wise over the 157B tokens (still cosine decay) instead of step-wise over the total number of steps. We describe the underlying rationale in Appendix A.2.

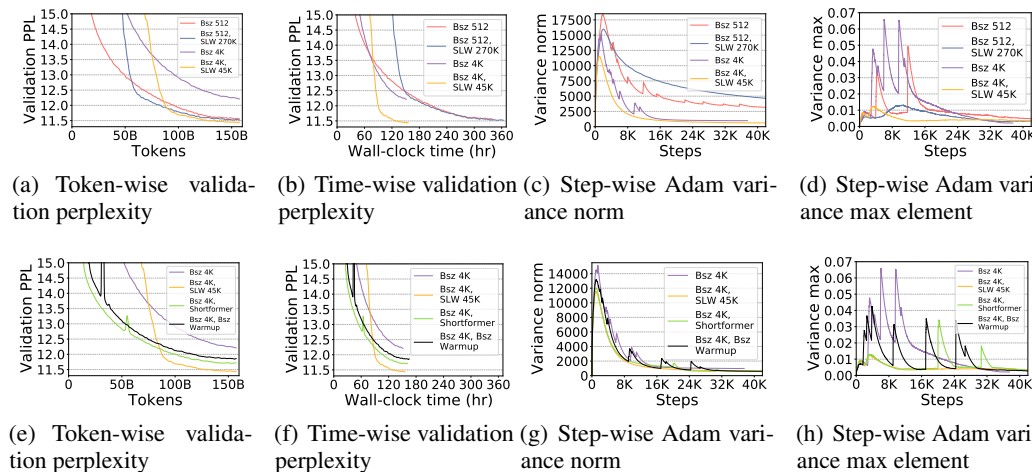

(a) Token-wise valida-
tion perplexity

(b) Time-wise validation
perplexity

(c) Step-wise Adam vari-
ance norm

(d) Step-wise Adam vari-
ance max element

(e) Token-wise valida-
tion perplexity

(f) Time-wise validation
perplexity

(g) Step-wise Adam vari-
ance norm

(h) Step-wise Adam vari-
ance max element

Figure 4: Validation perplexity and Adam variance norm/max element during GPT-2 1.5B seqlen 1K
pre-training, comparing the baseline and proposed work (SLW) under different batch sizes/LR. Also
compare with related works ("Shortformer" [30] and "Bsz Warmup" [6]) at 2nd row. Each row of
subfigures share the same legend ("SLW 45K" means our work with $T$=45K steps).

Based on the following observations, we demonstrate that our approach resolves the dilemma and
simultaneously improves the stability and efficiency. We will mainly present the GPT-2 1.5B results
and leave some GPT-2 117M results in Appendix.

**Significant stability gain:** In Section 3 Table 1 we discussed how we measure the training instability
based on the "loss ratio" metric, which shows that the baseline becomes less stable under larger
model size/batch size/learning rate/sequence length. Comparing with baseline and proposed work in
this table shows that our work reduces this instability measurement to zero in all cases, together with
max ratio close to 1.0 (no spike). This demonstrates the significant stability gain by our method.

**Faster token-wise and time-wise convergence:** Figure 4(a) and 4(b) present the validation perplexity
curves during GPT-2 1.5B seqlen 1K pre-training, comparing baseline and our approach. When the
batch size increases from 512 to 4K for baseline, the time-wise convergence becomes faster but the
token-wise convergence becomes slower and poorer. On the other hand, our approach at batch size
4K provides faster and better convergence both token-wise and time-wise comparing with the best
baseline curve in each case. The shape of SLW's curves is different (worse than baseline in early
stage) because SLW cases initially only learn from shorter sequences, which limit the validation
perplexity it can reach (since validation data is always full-length). On the other hand, when SLW
cases start to learn from longer sequences, the validation perplexity drops faster than baseline (and
related works) and eventually surpasses them.

Our approach with batch size 512 provides smaller convergence speedup because (1) Baseline
with batch size 512 has less instability issue, limiting the gain from the proposed approach; (2)
At batch size 512 the communication overhead is very high, and our approach takes more steps
(i.e., communication rounds) than baseline to reach the same 157B training tokens. This extra
communication cost "cancelled" part of the time saving from our approach. For GPT-2 117M, our
approach provides similar token-wise and time-wise convergence speedup (Appendix A.4).

**Advancing cost-quality Pareto curve:** In Section 3 Table 2 we discussed about baseline's zero-shot
evaluation results. For proposed work eval results in this table, we present them in two ways: one
evaluated at the earliest checkpoint that provides better eval results than baseline (batch size 512 and
seqlen 1K); the other one evaluated at the end of full training. Results show that our approach is able
to advance the cost-quality Pareto curve in two ways: (1) To reach the same eval result quality as
baseline, our approach reduces the required number of pre-training tokens and wall clock time by up
to 2.2x and 3.7x, respectively; (2) Under the same 157B training tokens, our approach can further
improve the eval result quality. In (1) the time-wise saving is higher than the token-wise because
(a) For each Transformer block, the self-attention and intermediate layers have time complexity of
$O(B \times L^2 \times H)$ and $O(B \times L \times H^2)$, respectively[6]. The proposed method uses shorter sequences at

---

[6]$B, L, H$ represent batch size, sequence length, hidden size.

the beginning, reducing the time complexity *quadratically* for the self-attention sub-layer and linearly for the intermediate sub-layer of Transformer blocks; (b) By enabling stable training at larger batch size, our approach achieves additional time-wise saving by reducing the communication overhead.

**Variance reduction helps stabilize training:** In Section 3 we discussed the strong correlation between training instability and gradient variance norm/max. Figure 4(c) and 4(d) demonstrate that proposed approach stabilizes training and reduces both the Adam variance norm and the variance max element. Importantly, it avoids all the spikes of the variance max element, which all happen to be where the baseline has training loss spikes. One may wonder why gradient clipping cannot help avoid these extreme gradient variance outliers. Although gradient clipping can avoid large gradient at every single step, it cannot avoid the gradient variance getting accumulated from multiple steps (Appendix A.3.2).

**Comparing with related works:** We now compare the proposed work with two related works on the most challenging "1.5B model + batch size 4K" case. The first work is the Shortformer where the first stage uses shorter sequences and the second stage uses full-length sequences [30]. Following the grid search in the paper, we use seqlen 128 for the first stage and set its duration at about half of the baseline duration (20K steps). The second work is the batch size warmup technique used by GPT-3 [6], where we set the starting batch size at 128 and then gradually increase it to 4K, and set the warmup duration same as the proposed work. Other training hyperparameters are unchanged. Figure 4(e) to 4(h) present the results. Both related works provide convergence speedup but it is less than our work. More importantly, they still have training instability issues. The Shortformer has an obvious training divergence at step 20K when the sequence length switches from 128 to 1K (the spike at 20K in Figure 4(h)). This shows when staying at the same shorter sequence length for too long, the model becomes heavily overfitted for that length which leads to divergence risk when/after switching to full length. Although both batch size warmup and our method reduce the number of tokens per batch in a similar fashion, batch size warmup does not provide any training stability benefit compared to the baseline. This indicates that providing the same number of shorter (simpler) sequences leads to better training stability than providing fewer number of same length (same difficulty) sequences. In addition, batch size warmup has a limitation that the batch size must be multiple of data-parallel size. On the other hand, for our method the sequence length only needs to be multiple of 8 to enable Tensor Core acceleration. These two limitations are different: for the proposed SLW method, the "multiple of 8" limitation is fixed and unrelated to data-parallel size. For batch size warmup it's a dynamic "multiple of data-parallel size" limitation: since nowadays pre-training tasks are performed on up to thousands of GPUs, the data-parallel size can easily go beyond 100, prohibiting flexible configuration of the method (or requires reducing the number of GPUs when using smaller batches, increasing the training clock time). Last but not least, both related works provide non-zero "loss ratio" in Table 1 and worse zero-shot evaluation results in Table 2.

### 5.2 GPT-3 experiments

For experiments replicating the GPT-3 125M model [6] using *the Pile* public dataset [13], first we reproduce the original GPT-3 training recipe: 300B training tokens, seqlen 2K, batch size 256 with batch size warmup (start with 16 then gradually increase to 256 in first 4B tokens), learning rate $6 \times 10^{-4}$ with a linear warmup of 375M tokens and a single cycle cosine decay over 260B tokens ($6 \times 10^{-5}$ min. learning rate)[7]. Then we explore an aggressive training scenario where only 30B tokens (10%) are allowed. This is because (1) GPT-3 paper admits that it has poor training sample efficiency and it sees much more text during pre-training than a human sees in the their lifetime [6, 23]. (2) There could exist cases where the total amount of data/computation resource is limited. We adjust several hyperparameters in this 30B-token training: 8x batch size (2K) for better training efficiency, learning rate decay reduced to 30B tokens (based on study that LR schedule should match total training tokens [14], warmup stays at 375M), min. learning rate reduced to 0 (based on recent study on GPT-3 [51]). For baseline we keep using 4B-token batch size warmup, but when our method is used ($seqlen_s = 72, T = 11.5K$ based on tuning) we disable it since both methods reduce tokens per batch. And for both cases we tune the learning rate and use the highest one that provides stable training, which is 30x ($1.8 \times 10^{-2}$) for baseline and 40x ($2.4 \times 10^{-2}$) for our method. All experiments are performed on 128 V100 GPUs.

Figure 5 and 6 present the training loss and gradient variance max for the GPT-3 pre-training experiments. When applying 40x learning rate to the baseline (batch size warmup), it quickly

---

[7]Different from GPT-2, GPT-3 uses token-based learning rate schedule and we follow it.

Table 4: Zero-shot evaluation of the trained GPT-3 125M models on 11 tasks used by the original GPT-3 work [6]. Per-task eval results in Appendix A.5.

| Case | Batch size | Training tokens | Training time | Average accuracy ↑ |
|------|-----------|-----------------|---------------|---------------------|
| 1: Original [6] | 256 | 300B | | 33.6 |
| 2: Baseline repro | 256 | 300B (1x) | 61Hr | 31.4 |
| 3: Baseline 30x LR | 2K | 30B (10x) | 7Hr (9x) | 29.8 (95%) |
| 4: SLW 40x LR | 2K | 30B (10x) | 3.5Hr (17x) | 31.1 (99%) |

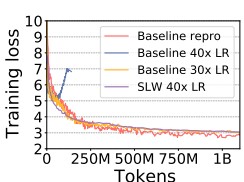

Figure 5: Training loss during GPT-3 125M pre-training (first 1B tokens).

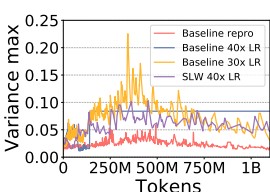

Figure 6: Gradient variance max during GPT-3 125M pre-training (first 1B tokens).

diverges and cannot continue to train due to NaN losses. The corresponding gradient variance max element becomes a flat line after divergence because the gradients on all dimensions are so large that all gradients get clipped including the max element. After lowering the learning rate to 30x, the baseline is able to finish the whole training, but it can only retain 95% of average zero-shot accuracy on 11 tasks (HellaSwag [53], LAMBADA [28], TriviaQA [15], WebQs [3], Winogrande [38], PIQA [4], ARC Challenge/Easy [10], ANLI R1/R2/R3 [27]) compared with the case that reproduces the original GPT-3 training recipe[8] as shown in Table 4. In contrast, our approach enables stable training with 40x learning rate, demonstrates lower gradient variance max outliers than baseline with 30x learning rate, retains 99% of the original training recipe's average zero-shot accuracy, and achieves 10x data saving and 17x time saving.[9] This demonstrates that the proposed method not only solves the stability-efficiency dilemma, but also opens a promising direction of significantly reducing total training cost in a different data efficiency dimension.

Finally, in Appendix A.6 we evaluate the proposed method on a larger GPT-3 1.3B model, including not only zero-shot but also few-shot evaluation. Results show that under the same 300B training tokens the proposed SLW method provides better average accuracy than the baseline for both zero-shot (from 41.6 to 41.9) and few-shot (from 44.8 to 45.3) tasks, demonstrating that the proposed method (in addition to the stability-efficiency benefit) can provide better accuracy performance. Similar to the original GPT-3, under few-shot prompts the average accuracy is better than zero-shot results for both models trained with baseline batch size warmup and proposed SLW method.

# 6 Limitation and Future Work

Although our paper discovers the correlation between long sequences and training instability (and proposes a method to alleviate the issue), the root cause of this causal relationship is not fully deciphered and would be an interesting future work direction. One assumption we have is that the shorter sequences are not necessarily easier but can be viewed as simpler examples since there are less context to embed. We think encoding shorter sequences (especially at the early training stage when weight is still relatively random) will generate less "noisy" gradients thus leading to higher stability.

This work presents correlation analysis on an empirical connection between training instability and gradient variance norm/max element, but it is not sufficient to prove a causal relationship, and training instability could be caused by other factors. Overall, our work is one preliminary attempt to study the topic of training instability in large-scale model training, and we hope it could inspire future works to further study this important issue in both theory and practice.

# 7 Conclusion

This paper presents the Sequence Length Warmup method, which solves a stability-efficiency dilemma inside GPT-style model pre-training, an critical issue that hinders efficient language model pre-training as explained in our in-depth analysis. By enabling stable training on more aggressive training recipe, this method also motivates a new dimension of training cost reduction by improving the data efficiency, as demonstrated by the 10x data and 17x time saving in our GPT-3 experiments. We believe that the effectiveness, simplicity, and easy-to-use/tune make the proposed method a must-try for deep learning practitioners, and we hope this work could motivate more studies on improving training data efficiency.

---

[8]Our reproduced GPT-3 has 2.2 point lower average accuracy than the original GPT-3, which is because of the different training data and OpenAI employed many data processing techniques [6]

[9]We want to emphasize here that the SLW method "only retains 99% of accuracy" because this experiment is an extreme case: only 10% of original training data is used during training.

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
