

(a) 117M training loss    (b) 1.5B training loss    (c) 117M variance norm    (d) 1.5B variance norm

(e) 117M variance max    (f) 1.5B variance max    (g) Var norm correlation    (h) Var max correlation

Figure 7: Training loss, Adam variance norm/max element, and correlations between loss spikes and variance norm/max during GPT-2 pre-training (without the proposed method) under different model sizes, batch sizes (and LR), and sequence lengths.

# A    Appendix

## A.1    Zoom in of Figure 1

Figure 7 zoom in the first 30B token in main paper Figure 1, where the training is the most unstable.

## A.2    Learning rate decay for proposed approach

As discussed in main paper Section 5.1 GPT-2 experiments, proposed approach needs more training steps than baseline in order to reach the same 157B training tokens. This makes it necessary to modify the learning rate decay schedule for proposed approach. We first tried to increase the number of learning rate decay steps by half of the proposed approach's pacing function duration $T$ (since the proposed approach roughly needs $T/2$ additional steps to reach 157B tokens). However, we find that simply increasing decay steps still leads to faster learning rate decay than baseline. At last we change the learning rate decay to token-wise (same cosine decay over the 157B tokens) instead of step-wise. This is because for the proposed approach there are fewer tokens per step at the beginning. So even if we increase the LR decay steps, it still cannot avoid decaying faster token-wise at the beginning compared to baseline. As shown in Figure 8, the proposed approach with step-wise LR decay (with $T/2$ extra decay steps) has faster LR decay token-wise compared to baseline, which leads to a worse validation perplexity curve. On the other hand, the same proposed approach case with token-wise LR decay has the same token-wise LR decay schedule as baseline, which leads to better convergence.

## A.3    Additional analysis about training hyperparameters

In main paper Section 4 we demonstrate that proposed approach's two hyperparameters can be tuned with very low cost only running the very beginning of the training (the third hyperparameter, ending sequence length, does not require tuning since it will always be the full length). To understand more about how proposed approach affects the choice and tuning of normal training hyperparameters, this section provides additional analysis about learning rates and gradient clipping. Results demonstrate that (a) Compared to baseline, proposed approach requires less tuning effort on these hyperparameters to provide a stable training; (b) By enabling stable training on larger learning rates, proposed approach could provide better training efficiency and convergence (as demonstrated in main paper Section 5); (c) Tuning gradient clipping for baseline could not provide the same training stability as proposed approach.

### A.3.1    Learning rate

In Section 5.1 we demonstrate that proposed approach can provide stable and more efficient training at larger batch size and learning rate, where baseline suffers from training instability. We increased both batch size and learning rate at the same time because (a) Large-batch training is more efficient

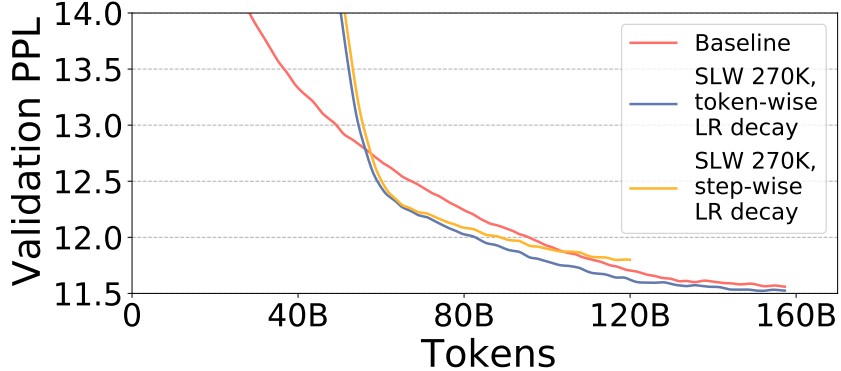

(a) Token-wise validation perplexity

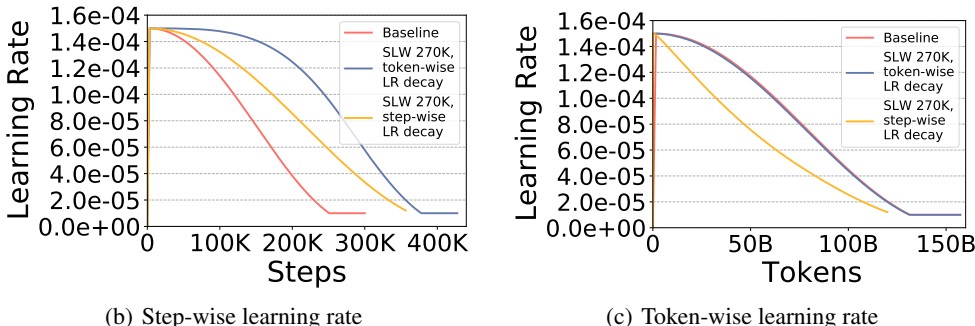

(b) Step-wise learning rate

(c) Token-wise learning rate

Figure 8: Validation perplexity and learning rate during GPT-2 1.5B seqlen 1K pre-training with batch size 512, comparing the baseline and proposed approach under different learning rate decay schedules. "SLW 270K" means proposed approach with $T$=270K steps.

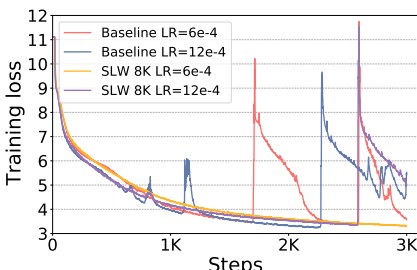

Figure 9: Step-wise training loss during GPT-2 1.5B seqlen 1K pre-training (first 3K steps only) with batch size 2K, seed 1236, and different learning rates for baseline and proposed approach ("SLW 8K" means proposed approach with $T$=8K steps).

for large-scale distributed training, so larger batch was necessary in our study (b) In order to maintain the same convergence speed, it is necessary to simultaneously increase the learning rate under larger batch size. A well-known rule of thumb is that the learning rate should at least increase by the square root of the batch size's increase ratio.

As a controlled experiment, here we perform additional studies about what if we keep the batch size the same and only tune learning rate for baseline and proposed approach. We do not consider the case of "same learning rate, different batch sizes" due to the reason (b) above. Table 5 presents the number of steps with training loss ratios (defined in main paper Section 3 as an indicative measurement of training instability) larger than 1.5 during GPT-2 1.5B seqlen 1K pre-training (first 3K steps only) with batch size 2K[11], 5 different seeds, and different learning rates for baseline and proposed approach. And Figure 9 illustrates some of the cases with seed 1236 to show how the loss spikes look

---

[11]Batch size 2K is used here because this analysis was performed at an early stage of this work, and we do not have enough resource to rerun the same analysis with batch size 4K.

Table 5: Number of steps with training loss ratios (defined in Section 3) larger than 1.5 during GPT-2 1.5B seqlen 1K pre-training (first 3K steps only) with batch size 2K, 5 different seeds, and different learning rates for baseline and proposed approach (SLW). Left/right number in each cell is for baseline/SLW, respectively.

| Baseline/SLW #loss ratio > 1.5 | LR = $1.5 \times 10^{-4}$ | LR = $3 \times 10^{-4}$ | LR = $6 \times 10^{-4}$ | LR = $12 \times 10^{-4}$ |
|---|---|---|---|---|
| Seed 1234 | 0/0 | 296/0 | 359/0 | 179/74 |
| Seed 1235 | 0/0 | 302/0 | 408/0 | 555/459 |
| Seed 1236 | 0/0 | 0/0 | 569/0 | 626/414 |
| Seed 1237 | 7/0 | 0/0 | 548/0 | 614/139 |
| Seed 1238 | 0/0 | 0/0 | 121/0 | 394/29 |
| Total | 7/0 | 598/0 | 2005/0 | 2368/1115 |

like. Results show that proposed approach provides stable training during this first 3K steps for all five seeds at learning rates up to $6 \times 10^{-4}$, while baseline with seed 1237 still has 7 large loss ratios at learning rate as low as $1.5 \times 10^{-4}$. At learning rate $12 \times 10^{-4}$ both cases have large loss ratios, but proposed approach reduces the frequency by 2.1x. This demonstrates that (a) Larger learning rates lead to higher training instability risk for both cases. (b) With the same amount of tuning effort, proposed approach has a higher probability of providing a stable training because of the wider range of learning rates it enables; (c) Since proposed approach enables stable training at larger learning rate, it could provide better and faster training convergence as shown in main paper Section 5.

### A.3.2 Gradient clipping

In main paper Section 5 we used gradient clipping at 1.0 (global gradient $l_2$ norm is clipped to 1.0) following the previous work [40]. Here we perform additional studies about what if we apply more gradient clipping to baseline. Figure 10(a) presents the training loss during GPT-2 1.5B seqlen 1K pre-training (first 5K steps only) with batch size 4K (the same hyperparameters as the second set in Section 3), comparing the baseline and proposed approach under different gradient clipping levels[12]. Results show that when applying more gradient clipping to baseline, the training has less and smaller loss spikes. And the Adam varaince norm is also reduced as shown in Figure 10(c).

However, more gradient clipping does not fully resolve the training instability issue. Even baseline with the lowest gradient clipping norm cannot avoid all training loss spikes, while proposed approach with default gradient clipping has no loss spike. As described in main paper, we believe that this is a limitation of common gradient clipping technique: Although gradient clipping can avoid too large gradient at every single step, it cannot avoid the gradient variance getting accumulated at certain dimensions (as shown in Figure 10(d)), especially for large batch sizes. Another concern about applying more gradient clipping is that the momentum norm is also reduced due to more clipping (Figure 10(b)). This indicates that when later the training reaches a more stable stage, more gradient clipping could hurt the convergence speed. On the other hand, proposed approach will not affect the convergence speed after the full sequence length is reached. Another thing to note is that proposed approach relies less on gradient clipping: at gradient clipping norm 1.0, baseline has 798 clippings in the first 5K steps while proposed approach has 628 clippings (21% less).

Overall, this analysis demonstrates that proposed approach requires less or no tuning on gradient clipping, while baseline still has training stability issue with more gradient clipping. It is possible that more complex and adaptive gradient/variance/activation clipping techniques could potentially achieve the same level of training stability as proposed approach. However, inventing and applying such techniques would require an effort no lower than the proposed approach, which is both easy to integrate and low-cost to tune.

### A.4 GPT-2 117M evaluation results

Figure 11 presents the validation perplexity and Adam variance norm/max element during GPT-2 117M pre-training, comparing the baseline and proposed work (SLW) under different batch sizes/LR and sequence lengths. Table 6 presents the zero-shot evaluation of the trained 117M models on

---

[12]We also tried less than 0.25 gradient clipping, which triggered a silent crash without error messages after around 100 steps. We did not have enough time to find the root cause, but it could be caused by the too extreme gradient clipping.

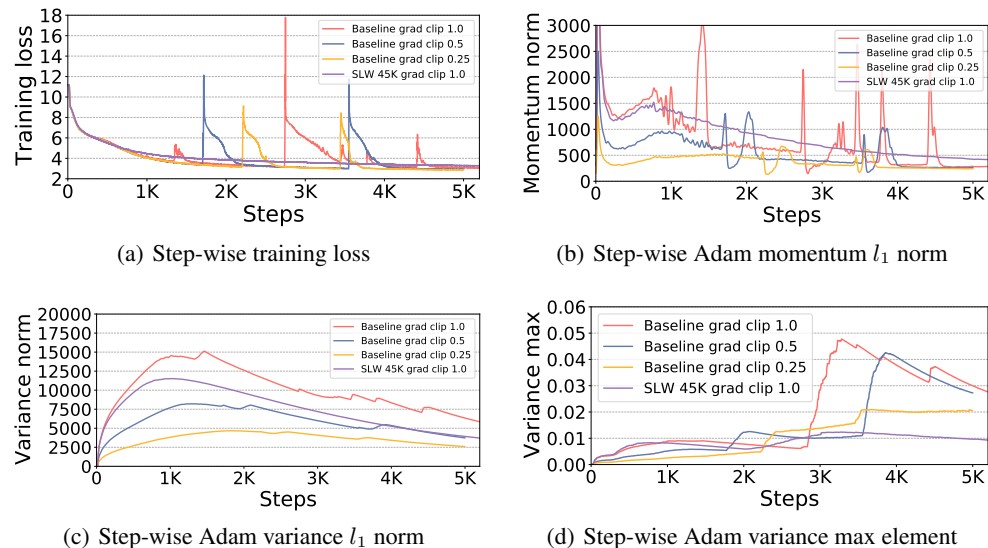

(a) Step-wise training loss

(b) Step-wise Adam momentum $l_1$ norm

(c) Step-wise Adam variance $l_1$ norm

(d) Step-wise Adam variance max element

Figure 10: Training loss, Adam momentum $l_1$ norm, and Adam variance $l_1$ norm/max element during GPT-2 1.5B seqlen 1K pre-training (first 5K steps only) with batch size 4K, comparing the baseline and proposed approach under different gradient clipping levels. Grad clip 1.0 indicates that the global gradient $l_2$ norm is clipped to 1.0. 'SLW 45K' means proposed approach with $T$=45K steps.

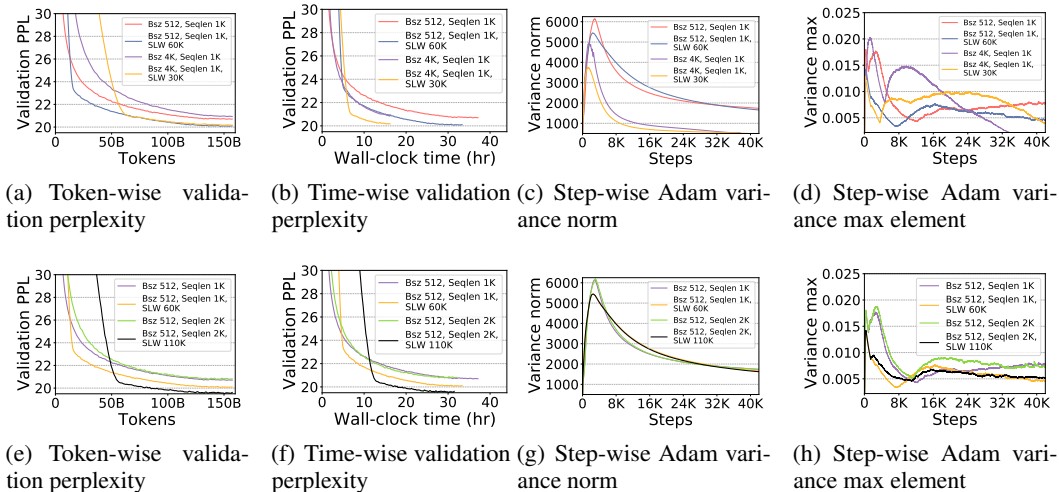

(a) Token-wise valida-tion perplexity

(b) Time-wise validation perplexity

(c) Step-wise Adam vari-ance norm

(d) Step-wise Adam vari-ance max element

(e) Token-wise valida-tion perplexity

(f) Time-wise validation perplexity

(g) Step-wise Adam vari-ance norm

(h) Step-wise Adam vari-ance max element

Figure 11: Validation perplexity and Adam variance norm/max element during GPT-2 117M pre-training, comparing the baseline and proposed work (SLW) under different batch sizes/LR and sequence lengths. "SLW 60K" means proposed work with $T$=60K steps.

the WikiText-103 and LAMBADA datasets for baseline and proposed work with different pacing function duration.

## A.5   GPT-3 125M evaluation results

Table 7 presents the zero-shot evaluation of the trained GPT-3 125M models on the 11 tasks used by the original GPT-3 work [6].

Table 6: Zero-shot evaluation of the GPT-2 117M models on the WikiText-103 and LAMBADA datasets, following the evaluation methodology from [40].

| Case | Pre-training parameters | Pre-training steps, tokens, time | Pre-training test perplexity ↓ | WikiText-103 perplexity ↓ | LAMBADA accuracy ↑ |
|---|---|---|---|---|---|
| 1: Baseline | bsz512-seqlen1K | 300K, 157B, 37Hr | 20.75 | 27.78 | 33.19% |
| 2: SLW 20K | bsz512-seqlen1K | 310K, 157B, 30Hr | 20.49 | 27.43 | **34.60%** |
| **3: SLW 60K** | bsz512-seqlen1K | 330K, 157B, 33Hr | **20.11** | 27.01 | 34.41% |
| 4: SLW 100K | bsz512-seqlen1K | 350K, 157B, 35Hr | 20.16 | **26.91** | 34.21% |
| 5: SLW 140K | bsz512-seqlen1K | 370K, 157B, 35Hr | 20.17 | 27.17 | 33.92% |
| 6: Baseline | bsz4K-seqlen1K | 37.5K, 157B, 16Hr | 20.99 | 28.09 | 32.54% |
| 7: SLW 10K | bsz4K-seqlen1K | 42.5K, 157B, 16Hr | 20.34 | 27.22 | 33.98% |
| 8: SLW 20K | bsz4K-seqlen1K | 47.5K, 157B, 16Hr | 20.25 | 27.13 | **34.54%** |
| **9: SLW 30K** | bsz4K-seqlen1K | 52.5K, 157B, 16Hr | **20.22** | 27.15 | 34.16% |
| 10: SLW 40K | bsz4K-seqlen1K | 57.5K, 157B, 16Hr | 20.26 | **27.11** | 33.53% |
| 13: Baseline | bsz512-seqlen2K | 150K, 157B, 32Hr | 20.87 | 28.19 | 32.99% |
| 15: SLW 70K | bsz512-seqlen2K | 185K, 157B, 31Hr | 19.82 | 26.04 | 33.46% |
| **17: SLW 110K** | bsz512-seqlen2K | 205K, 157B, 31Hr | **19.64** | 26.03 | **34.58%** |
| 18: SLW 150K | bsz512-seqlen2K | 215K, 157B, 32Hr | **19.64** | **25.99** | 33.32% |
| 15: SLW 190K | bsz512-seqlen2K | 245K, 157B, 33Hr | **19.64** | 26.09 | 33.09% |

Table 7: GPT-3 125M zero-shot evaluation results

| Case | Original [6] | Baseline repro | Baseline 30x LR | SLW 40x LR |
|---|---|---|---|---|
| Model size | 125M | 125M | 125M | 125M |
| Train tokens | 300B | 300B | 30B | 30B |
| Batch size | 256 | 256 | 2K | 2K |
| Bsz warmup | 4B | 4B | 4B | N/A |
| LR | 6e-4 | 6e-4 | 1.8e-2 | 2.4e-2 |
| min LR | 6e-5 | 6e-5 | 0 | 0 |
| LR warmup | 375M | 375M | 375M | 375M |
| LR decay | 260B | 260B | 30B | 30B |
| decay style | cosine | cosine | cosine | cosine |
| SLW | N/A | N/A | N/A | 11.5K steps |
| Avg. accuracy | 33.6 | 31.4 | 29.8 | 31.1 |
| (0) HellaSwag | 33.7 | 30.4 | 28.2 | 28.9 |
| (1) LAMBADA | 42.7 | 39.3 | 30.4 | 34.2 |
| (2) TriviaQA | 4.15 | 1.72 | 0.76 | 1.45 |
| (3) WebQs | 1.77 | 0.197 | 0 | 0.394 |
| (4) Winogrande | 52.0 | 49.3 | 50.9 | 51.9 |
| (5) PIQA | 64.6 | 61.9 | 59.8 | 62.7 |
| (6) ARC Challenge | 26.6 | 23.3 | 21.7 | 22.3 |
| (7) ARC Easy | 43.6 | 39.9 | 36.0 | 39.1 |
| (8) ANLI R1 | 33.4 | 32.8 | 33.1 | 33.4 |
| (9) ANLI R2 | 33.2 | 33.3 | 33.3 | 33.6 |
| (10) ANLI R3 | 33.6 | 33.3 | 33.2 | 34.7 |

## A.6 GPT-3 1.3B evaluation results

In this section we evaluate the proposed SLW method on the larger GPT-3 1.3B model. Compared to the GPT-3 125M evaluation in main paper section 5.2 there are two differences on the setup: (1) The GPT-3 125M evaluation aims to explore whether the proposed method can retain the accuracy performance while greatly reducing the training tokens, while this GPT-3 1.3B evaluation aims to explore that, under same amount of training tokens, does proposed method provides better training stability and better accuracy performance. (2) To improve the training data quality, for GPT-3 1.3B pre-training we added two additional sub-datasets (CC-Stories [45] and RealNews [54]), together with additional data cleaning on all data following the process in [42].

Similar to previous experiments, we test two set of hyperparameters on both baseline and proposed method: The first set follows the original GPT-3 setup: 300B training tokens, seqlen 2K, batch size 512 (baseline case includes batch size warmup that starts with 16 then gradually increase to 512 in first 8B tokens), learning rate $2 \times 10^{-4}$ with a linear warmup of 375M tokens and a single cycle cosine decay over 260B tokens ($2 \times 10^{-5}$ min. learning rate). The second set changes the batch size to 4K (8x) and learning rate to $8 \times 10^{-4}$ (4x).

Table 8: GPT-3 1.3B zero-shot evaluation results

| Case | Original [6] | Baseline repro | SLW 8x Bsz |
|---|---|---|---|
| Model size | 1.3B | 1.3B | 1.3B |
| Train tokens | 300B | 300B | 300B |
| Batch size | 512 | 512 | 4K |
| Bsz warmup | 8B | 8B | N/A |
| LR | 2e-4 | 2e-4 | 8e-4 |
| min LR | 2e-5 | 2e-5 | 2e-5 |
| LR warmup | 375M | 375M | 375M |
| LR decay | 260B | 260B | 260B |
| decay style | cosine | cosine | cosine |
| SLW | N/A | N/A | 11K steps |
| Avg. accuracy | 44.4 | 41.6 | 41.9 |
| (0) LAMBADA | 63.6 | 63.7 | 65.0 |
| (1) TriviaQA | 19.7 | 10.1 | 11.3 |
| (2) WebQs | 4.63 | 3.25 | 2.36 |
| (3) PIQA | 75.1 | 73.4 | 73.8 |
| (4) RACE-h | 40.9 | 35.6 | 37.1 |
| (5) BoolQ | 62.4 | 63.4 | 61.8 |

The baseline case only enables stable training on the first set of hyperparameters. Under larger batch size and learning rate, a training divergence (similar to main paper Figure 5 blue line) happened and the training cannot continue. On the other hand, the proposed SLW method is able to provide stable training under 8x larger batch size and 4x larger learning rate. Under the same number of training tokens, the 8x larger batch size leads to better training efficiency and 2x training time speedup, similar to what we obserbe in GPT-2 pre-training (main paper Table 2 case 10 vs. 15). This demonstrate the stability-efficiency benefit of the proposed method.

In addition, Table 8 and 9 present the zero-shot and few-shot evaluations of the trained GPT-3 1.3B models on 6 tasks used by the original GPT-3 work [6]: LAMBADA [28], TriviaQA [15], WebQs [3], PIQA [4], RACE-h [20], BoolQ [47]. Results show that similar to the original GPT-3, under few-shot prompts the average accuracy is better than zero-shot results for both models trained with baseline batch size warmup (from 41.6 to 44.8) and proposed SLW method (from 41.9 to 45.3).[13] The change on each task also follows the same pattern: TriviaQA and WebQs accuracy improve a lot under few-shot; PIQA, RACE-h, and BoolQ have similar accuracy under zero and few-shot; LAMBADA accuracy becomes worse under few-shot. More importantly, under the same 300B training tokens the proposed SLW method provides better average accuracy (zero-shot from 41.6 to 41.9, few-shot from 44.8 to 45.3) than the baseline, demonstrating that the proposed method (in addition to the stability-efficiency benefit) is able to provide better accuracy performance.

---

[13]Similar to main paper section 5.2, our reproduced GPT-3 baseline has 2.9/3.3 point lower average zero/few-shot accuracy than the original GPT-3, which is because of the different training data and OpenAI employed special data processing techniques [6]

Table 9: GPT-3 1.3B few-shot evaluation results. k denotes the number of shots following the original GPT-3 work [6].

| Case | Original [6] | Baseline repro | SLW 8x Bsz |
|---|---|---|---|
| Model size | 1.3B | 1.3B | 1.3B |
| Train tokens | 300B | 300B | 300B |
| Batch size | 512 | 512 | 4K |
| Bsz warmup | 8B | 8B | N/A |
| LR | 2e-4 | 2e-4 | 8e-4 |
| min LR | 2e-5 | 2e-5 | 2e-5 |
| LR warmup | 375M | 375M | 375M |
| LR decay | 260B | 260B | 260B |
| decay style | cosine | cosine | cosine |
| SLW | N/A | N/A | 11K steps |
| Avg. accuracy | 48.1 | 44.8 | 45.3 |
| (0) LAMBADA (k=15) | 57.0 | 58.8 | 59.7 |
| (1) TriviaQA (k=64) | 32.1 | 19.2 | 19.0 |
| (2) WebQs (k=64) | 19.6 | 18.4 | 19.4 |
| (3) PIQA (k=50) | 74.3 | 74.2 | 72.8 |
| (4) RACE-h (k=10) | 41.4 | 35.0 | 37.6 |
| (5) BoolQ (k=32) | 64.1 | 63.2 | 63.2 |