# OpenReview forum: "The Stability-Efficiency Dilemma: Investigating Sequence Length Warmup for Training GPT Models"
_NeurIPS.cc/2022/Conference — NeurIPS 2022 Accept_

### Official Review · Reviewer_c8jb · 2022-07-06

**Rating:** 6
**Confidence:** 4
**Soundness:** 3 good
**Presentation:** 3 good
**Contribution:** 3 good

**Summary:**

This work introduces the Sequence Length Warmup (SLW) method: a simple but demonstrably effective technique for improving training stability of GPT models. This paper contributes some characterization of the instability problem, possible connections to gradient variance, and empirical evidence that SLW can sidestep this issue. The main body of the paper elaborates on implementation details and includes various comparisons to quantify the specific cost/quality improvements offered by SLW. The paper concludes with experiments with GPT-3, which further demonstrate the technique.

**Questions:**

Is it fair to use the same distributed set-up for comparing the 512 vs. 4k batch size settings? Presumably one easy way to reduce communication cost would be to use fewer nodes with the smaller batch size. From my understanding, all jobs were run with the same hardware, which might mean that the 512 baseline is slower than it needs to be. I'm not sure about this potential criticism, so I would appreciate it if the authors could explain why this is a fair comparison in terms of efficiency.

Have you done any fine-tuning experiments to confirm that the benefits to zero-shot learning would also apply to other fine-tuning settings? If there is suitable prior work to establish this, it would be helpful to mention somewhere in the paper.

**Limitations:**

The paper doesn't have a "societal impact" section but I think that is OK here.

**Strengths And Weaknesses:**

# Strengths:
- Within the experimental scope of the paper, the results show clear improvements, particularly in gaining access to more "aggressive" (and more efficient) training settings.
- The authors make good use of a simple loss spike and loss ratio metric for quantifying instability. Through this lens, SLW has a dramatic reduction on instability as it generally reduces this quantity to 0. (But see weaknesses below)
- SLW is simple to implement and does not seem to require prohibitive hyperparameter tuning. (Although, the tuning heuristic suggested by the authors is pretty vague -- e.g., "significant validation perplexity fluctuation" is unclear.)
- The comparison between SLW and batch size warmup is interesting.

# Weaknesses:
- The empirical connection between training instability and gradient variance is not particularly convincing (Fig 1, Table 3). This connection is limited to a correlation analysis, but it is easy to doubt any causal implications. I expect that there are many arbitrary measurements of the training history that statistically significantly correlate with gradient variance.
- The best metric for showing that SLW reduces instability is loss spikes. However, there isn't any clear evidence that loss spikes interfere with training quality. I'm willing to give the benefit of the doubt here, but it would at least make things cleaner if some analysis were provided that would better motivate the goal of preventing loss spikes.
- The narrative of this paper is hard to follow. Ultimately, this weakens the characterization of the "stability-efficiency dilemma." In sum, it seems safe to say that it reflects some poorly understood interplay between batch size, learning rate, gradient variance, and sequence length, but all that this paper really teaches us is that we can make it less of a dilemma by doing SLW. In many ways, that's a perfectly useful conclusion, but it does limit the *understanding* we gain of the phenomenon. As a more concrete example, the motivation for SLW seems to follow the observation that training instability requires sequences longer than 128. The authors leverage that observation into a clearly useful technique, but, in all honesty, all the surrounding analysis and conjecture is very *ad hoc* and not very convincing. It would have been far more valuable to use some of this paper space to dig into sequence length as a driver of instability. It's too bad that we're left with no new knowledge other than that it is involved and SLW helps.

# Comments:
- In Section 2, it would be helpful to unpack some of the prior works referenced in the *Curriculum Learning* paragraph, especially those where "CL was explored and verified for NLP one-stage and fine-tuning tasks." The current phrasing is too vague and fails to provide proper context for this paper/work.
- You write: "Baseline with batch size 4K is less stable than baseline with batch size 512, indicating that larger batch sizes could lead to more training instability risks." This seems like an unwarranted claim since batch size is confounded with learning rate here. The authors should clarify this.
- In Figure 2, it would be helpful to annotate the 10% of time where the yellow curve has 1k sequence length, especially since the main text says that these portions of training tend to trigger the instability. Also, please double check that the steps listed in line 213 ("900, 1800, 2700...") are accurate.
- The figures are often hard to read because so much visual space is being used for labels under subfigures, axis numbers, etc. This will not affect my evaluation but it would be good to improve figure quality before finalizing the paper.

---

> ### Author Response · Authors · 2022-07-29
> **Response part 1**
>
> Thank you for your comments and below are our replies.
>
> <Comment 1> "Although, the tuning heuristic suggested by the authors is pretty vague -- e.g., "significant validation perplexity fluctuation" is unclear."
>
> <Reply 1> We agree and will add a quantitative definition of the "significant validation perplexity fluctuation". Preliminary analysis shows that a quantitative threshold would be "whether the perplexity value becomes larger than 1.3x of the previous best perplexity".
>
> <Comment 2> "The empirical connection between training instability and gradient variance is not particularly convincing (Fig 1, Table 3). This connection is limited to a correlation analysis, but it is easy to doubt any causal implications. I expect that there are many arbitrary measurements of the training history that statistically significantly correlate with gradient variance."
>
> <Reply 2> We agree that the correlation analysis is not sufficient to prove a causal relationship, and training instability could be caused by other factors. In the final version, we will add a discussion of this limitation and potential future work.
>
> <Comment 3> "The best metric for showing that SLW reduces instability is loss spikes. However, there isn't any clear evidence that loss spikes interfere with training quality. I'm willing to give the benefit of the doubt here, but it would at least make things cleaner if some analysis were provided that would better motivate the goal of preventing loss spikes."
>
> <Reply 3> In the paper, our experiments show that training instabilities lead to loss spikes. Some of these spikes lead to irrecoverable divergence (e.g., Figure 5). Others can recover, but it requires a non-trivial number of training steps to recover the training loss back to the normal level (e.g., Figure 1(b)). These training "detours" slow down the learning and eventually lead to worse accuracy performance under the same training budget (e.g., Table 2 case 13 vs case 10), or requires significantly more training iterations to reach a potentially similar model quality. We will add this discussion in the final version.
>
> <Comment 4> "The narrative of this paper is hard to follow. Ultimately, this weakens the characterization of the "stability-efficiency dilemma." In sum, it seems safe to say that it reflects some poorly understood interplay between batch size, learning rate, gradient variance, and sequence length, but all that this paper really teaches us is that we can make it less of a dilemma by doing SLW. In many ways, that's a perfectly useful conclusion, but it does limit the understanding we gain of the phenomenon. As a more concrete example, the motivation for SLW seems to follow the observation that training instability requires sequences longer than 128. The authors leverage that observation into a clearly useful technique, but, in all honesty, all the surrounding analysis and conjecture is very ad hoc and not very convincing. It would have been far more valuable to use some of this paper space to dig into sequence length as a driver of instability. It's too bad that we're left with no new knowledge other than that it is involved and SLW helps."
>
> <Reply 4> To understand the relationship between sequence length and instability, in the paper we performed some analysis (e.g., figure 2 and the paragraph on page 5) to show that long sequence length, especially during the early stage, is harmful to stability. We agree that additional investigation would be helpful, and would love to hear any suggestions for additional analysis from the reviewer.
>
> <Comment 5> "In Section 2, it would be helpful to unpack some of the prior works referenced in the Curriculum Learning paragraph, especially those where "CL was explored and verified for NLP one-stage and fine-tuning tasks." The current phrasing is too vague and fails to provide proper context for this paper/work."
>
> <Reply 5> We agree and will extend this related work discussion in the final version.
>
> <Comment 6> "You write: "Baseline with batch size 4K is less stable than baseline with batch size 512, indicating that larger batch sizes could lead to more training instability risks." This seems like an unwarranted claim since batch size is confounded with learning rate here. The authors should clarify this."
>
> <Reply 6> Yes, what we actually mean is that when increasing the batch size by K, the learning rate often needs to be scaled accordingly (e.g., sqrt(K)) to achieve a similar convergence speed, and this combination leads to a higher chance of training instability. We will fix this statement in the final version.

---

> ### Author Response · Authors · 2022-07-29
> **Response part 2**
>
> <Comment 7> "In Figure 2, it would be helpful to annotate the 10\% of time where the yellow curve has 1k sequence length, especially since the main text says that these portions of training tend to trigger the instability. Also, please double check that the steps listed in line 213 ("900, 1800, 2700...") are accurate."
>
> <Reply 7> We agree and we had submitted a revision where in figure 2 we use the cyan background to represent the steps that have full-length sequences. And yes the "900, 1800, 2700..." is a typo: it's actually "900, 1900, 2900..." and we fixed it in the revision.
>
> <Comment 8> "The figures are often hard to read because so much visual space is being used for labels under subfigures, axis numbers, etc. This will not affect my evaluation but it would be good to improve figure quality before finalizing the paper."
>
> <Reply 8> We agree and will improve the readability of figures in the final version.
>
> <Comment 9> "Is it fair to use the same distributed set-up for comparing the 512 vs. 4k batch size settings? Presumably one easy way to reduce communication cost would be to use fewer nodes with the smaller batch size. From my understanding, all jobs were run with the same hardware, which might mean that the 512 baseline is slower than it needs to be. I'm not sure about this potential criticism, so I would appreciate it if the authors could explain why this is a fair comparison in terms of efficiency."
>
> <Reply 9> It is true that under fewer nodes, smaller batch sizes can also achieve good computation efficiency. However, in practice, the goal of a training task is usually "given a fixed number of hardwares, how to train the model in the fastest wall clock time". And given the increasing model sizes, pre-training on hundreds of GPUs is not uncommon. Thus we believe that using the same hardware resources is a fair comparison. We will add this clarification in the final version.
>
> <Comment 10> "Have you done any fine-tuning experiments to confirm that the benefits to zero-shot learning would also apply to other fine-tuning settings? If there is suitable prior work to establish this, it would be helpful to mention somewhere in the paper."
>
> <Reply 10> In related GPT-style pre-training works, usually zero-shot and few-shot evaluations are used to measure model quality instead of fine-tuning (e.g., the GPT-3 paper "language models are few shot learners"). Part of the reason is that it becomes expensive to fine-tune such large models on many downstream tasks. Thus in the original submission we focused on zero-shot evaluation, and after submission, we performed some studies of few-shot evaluation on a larger GPT-3 1.3B model, and we have just added it to the revised supplementary material A.5. Results show that similar to the original GPT-3, under few-shot prompts the average accuracy is better than zero-shot results for both models trained with baseline batch size warmup (from 41.6 to 44.8) and proposed SLW method (from 41.9 to 45.3). In addition, the proposed SLW method provides better average accuracy (zero-shot from 41.6 to 41.9, few-shot from 44.8 to 45.3) than the baseline.

---

### Official Review · Reviewer_S5Qt · 2022-07-07

**Rating:** 6
**Confidence:** 3
**Soundness:** 3 good
**Presentation:** 2 fair
**Contribution:** 3 good

**Summary:**

In this work, the authors address the difficulty of training large language models with a novel strategy: sequence length warmup. The authors note that transformer training requires limited batch size and learning rate, and that methods which allow both to be increased can improve training efficiency. Their proposal is to use a training strategy which uses short sequence length examples to start, and gradually brings in longer examples.

The authors argue that though increasing batch size decreases the overall norm of the gradient variance estimation vector used by ADAM, but increases the largest element - which may cause training instability. They show that sequences with longer lengths also display the same phenomenology. They correlate large values of this largest element with issues with the stability of training.

The authors then propose a strategy to combat these instabilities: sequence length warmup. During the warmup phase, the maximum sequence length of seen examples is slowly increased. This leads to better training stability at larger batch size, and overall quicker convergence (in terms of number of steps) as well as more efficient training (in terms of number of tokens seen). In particular the sequence length warmup with a linear rampup of sequence length outperforms the shortformer, which has a step-function increase in sequence length.

**Questions:**

One issue with the analysis in Section 3 is that both the batch size AND the learning rate have been increased simultaneously. What happens when large batch size is used with a different learning rate? It seems like you may be able to better trade efficiency and stability with simultaneous optimization of both parameters.

With regards to the analysis of the norm/max element of the variance vector: the fact that the max element increases with batch size runs counter to what one would expect from distributions with finite moments - but is a known property of long-tailed distributions (e.g. those with power law tails). Is there a way to see this directly? E.g. by computing moments/distributions of the gradients near initialization/during training?

Recent work has shown the importance of the curvature of the loss landscape on optimization, particular in the early stages of learning in the transformer architecture:

https://openreview.net/forum?id=OcKMT-36vUs

Does sequence length warmup improve the conditioning of the Hessian?

Is there an explanation for the significantly different shapes of the learning curves in Figure 4 (particualarly (e) and (f)) of the SLW method as compared to the other methods? It's interesting that the curves are qualitatively different.

**Limitations:**

Adequately addressed.

**Strengths And Weaknesses:**

The analysis of the failure modes of the transformer training with respect to batch size and sequence length was quite original and reasonably clearly presented. The results with regards to sequence length are particularly compelling. One criticism is that figures like Figure 1 are hard to parse - both because of the total number of plots, as well as their size. In addition, details of the pre-warmup quantities is of theoretical and practical interest and is unclear from the plot.

However one weakness of the analysis is that batch size and learning rate were increased simultaneously. It leaves open the question of whether or not a better learning rate tuning would allow for the use of a larger batch size, while still maintaining some of the efficiency gains. This is also where a zoomed in look at the dynamics of the quantities early in training would be important; larger batch size should have smaller variability with equal parameters, so something in the early learning dynamics pushes the model to have more variance.

It is not clear to me that spikes in the max variance is itself is an issue; with an ADAM optimizer and/or gradient clipping, updates in these large gradient directions will be suppressed. In addition, a larger overall norm is not intrinsically a problem if paired with the right learning rate schedule/an adaptive optimizer like ADAM. It does seem though that those quantities may be indicative of larger issues with  the optimization/loss landscape progression during learning.

Regardless of the issues with the analysis of the stability-efficiency tradeoff, the sequence length warmup algorithm seems to be of great significance - leading to improved performance in terms of compute and final attainable perplexity. The overall sequence length warmup algorithm is very similar to the cited Shortformer work, though the small modifications seem to have provided significant improvements. Figure 3 was particularly interesting - showing the tradeoff between learning curves in terms of number of total steps versus number of total tokens seen.

The discussion of changing the cost-quality Pareto frontier was also interesting, but felt incomplete; the paper could be improved with more explicit exploration of this frontier.

---

> ### Author Response · Authors · 2022-07-29
> **Response part 1**
>
> Thank you for your comments and below are our replies.
>
> <Comment 1> "One criticism is that figures like Figure 1 are hard to parse - both because of the total number of plots, as well as their size. In addition, details of the pre-warmup quantities is of theoretical and practical interest and is unclear from the plot."
>
> <Reply 1> Figure 1 has many plots because we were trying to illustrate various information related to the stability-efficiency dilemma: loss spikes and their relationship to gradient norm/max element. We agree and will improve the readability of figures in the final version. On the other hand, we are not very sure about the meaning of the "details of the pre-warmup quantities" you mentioned. Do you mean the training experiments without the proposed sequence length warmup method (i.e., the baseline cases), or something else?
>
> <Comment 2> "However one weakness of the analysis is that batch size and learning rate were increased simultaneously. It leaves open the question of whether or not a better learning rate tuning would allow for the use of a larger batch size, while still maintaining some of the efficiency gains. This is also where a zoomed in look at the dynamics of the quantities early in training would be important; larger batch size should have smaller variability with equal parameters, so something in the early learning dynamics pushes the model to have more variance."
>
> "One issue with the analysis in Section 3 is that both the batch size AND the learning rate have been increased simultaneously. What happens when large batch size is used with a different learning rate? It seems like you may be able to better trade efficiency and stability with simultaneous optimization of both parameters."
>
> <Reply 2> We agree that experiments that tune learning rates under the same batch size would be helpful, and actually we did conduct this kind of study and included it in appendix A.2.1. Results show that under the same large batch size, the proposed SLW method enables stable training at larger learning rates, which otherwise would lead to training divergence. It also alleviates the tuning cost and enables a better stability-efficiency trade-off than the baseline. In the final version, we will include a clearer reference to this appendix in the main paper.
>
> <Comment 3> "It is not clear to me that spikes in the max variance is itself is an issue; with an ADAM optimizer and/or gradient clipping, updates in these large gradient directions will be suppressed. In addition, a larger overall norm is not intrinsically a problem if paired with the right learning rate schedule/an adaptive optimizer like ADAM. It does seem though that those quantities may be indicative of larger issues with the optimization/loss landscape progression during learning."
>
> <Reply 3> All the experiments use Adam optimizer and we agree that experiments about gradient clipping would be helpful, and actually we did conduct this kind of study and included it in appendix A.2.2. Results show that the proposed approach requires less or no tuning on gradient clipping, while baseline still has training stability issue with various gradient clipping values. In the final version, we will include a clearer reference to this appendix in the main paper.
>
> <Comment 4> "The discussion of changing the cost-quality Pareto frontier was also interesting, but felt incomplete; the paper could be improved with more explicit exploration of this frontier."
>
> <Reply 4> We agree that the study could be extended, though exploring the full Pareto frontier is very resource and time-consuming for the pre-training tasks. In the final version, we will extend it as much as resource permits.
>
> <Comment 5> "With regards to the analysis of the norm/max element of the variance vector: the fact that the max element increases with batch size runs counter to what one would expect from distributions with finite moments - but is a known property of long-tailed distributions (e.g. those with power law tails). Is there a way to see this directly? E.g. by computing moments/distributions of the gradients near initialization/during training?"
>
> <Reply 5> We thank the reviewer for this good suggestion. We think the distribution of gradients/moments follows a normal distribution with a long tail, especially for large batch sizes because larger batch has a higher chance of hitting outliers. We will try to add this kind of analysis and some illustrations in the final version.

---

> > ### Comment · Reviewer_S5Qt · 2022-08-02
> > **Responses to responses**
> >
> > Apologies for my lack of clarity in comment 1. By "details of the pre-warmup quantities", what I meant to say is that the values of the plots in the warmup phase are of particular interest, and there should be a way to clearly see them (via an inset, or log-scaling of the x-axis). Currently they are hart to parse.
> >
> > Re: reply 5: It would be good to have some distributional measures in the final text. I'd also like to point out that "normal distribution" and "long tail" are contradictory - would be great to know which one it is!

---

> > > ### Author Response · Authors · 2022-08-03
> > > **Response**
> > >
> > > Thank you for clarifying the "pre-warmup quantities". The learning rate warmup for GPT-2 pre-training is the first 3K steps, which is the first 1%/8% of training under batch size 512/4K. This phase does have some distinct properties, for example the gradient variance norm/max element continuously increase during warmup. We will illustrate and discuss about this in the final version. One solution we can think of is adding another set of plots (to the appendix if there is not enough space) only showing the initial 10% of the training.
> > >
> > > About the distributional measures, we agree that "normal distribution" and "long tail" are contradictory, and our conjecture is that when the batch size is larger it will look more like a "long tail", and vice versa. We will work on this analysis and include it in the final version.

---

> ### Author Response · Authors · 2022-07-29
> **Response part 2**
>
> <Comment 6> "Recent work has shown the importance of the curvature of the loss landscape on optimization, particular in the early stages of learning in the transformer architecture: https://openreview.net/forum?id=OcKMT-36vUs. Does sequence length warmup improve the conditioning of the Hessian?"
>
> <Reply 6> We agree and there are works that demonstrate curriculum learning helps smooth the loss landscape (e.g., SAT: Improving Adversarial Training via Curriculum-Based Loss Smoothing, Sitawarin et al, https://arxiv.org/abs/2003.09347). Given that SLW follows the same paradigm of curriculum learning, we believe SLW also improves the Hessian of the model. We will try to add a Hessian analysis in the final version or future work.
>
> <Comment 7> "Is there an explanation for the significantly different shapes of the learning curves in Figure 4 (particualarly (e) and (f)) of the SLW method as compared to the other methods? It's interesting that the curves are qualitatively different."
>
> <Reply 7> The shape of SLW's curves is different because SLW cases initially only learn from shorter sequences which "limit" the validation perplexity it can reach (since validation data is always full-length). On the other hand, when SLW cases start to learn from longer sequences, the validation perplexity drops faster than baseline and alternative approaches and eventually surpasses them. We will add this discussion in the final version.

---

### Official Review · Reviewer_iCLQ · 2022-07-09

**Rating:** 5
**Confidence:** 5
**Soundness:** 2 fair
**Presentation:** 3 good
**Contribution:** 2 fair

**Summary:**

This paper proposed a simple and effective method, "sequence length warmup" (SLW), to improve the training stability and efficiency, in GPT model pre-training.

In different experiment configurations, this paper reveals the "stability-efficiency dilemma for GPT model pre-training", and a strong correlation between training instability and extreme values of gradient variance.

The experiments are conducted on the pre-training of GPT-2 (117 M and 1.5 B parameters) and GPT-3 (125M parameters): SLW reduces the required number of training tokens and wall clock time by up to 2.2x and 3.7x. For GPT-3 (125M parameters), when limited to 10% of data available, 8X larger batch size and 40x larger learning rate, SLW can retraining 99% of the zero-shot accuracy, in 12x less time. In the same setting, baseline can retrain 95% of the zero-shot accuracy after lowering learning rate to 30x.

**Questions:**

* How about the few-shot performance of the pre-trained model using SLW? Adding the few-shot results (no matter higher or lower), would give us a more clear picture of how well SLW is. Because just like in the title of GPT-3 paper, "Language models are few-shot learners", we need to know how well SLW is in the few-shot settings, if SLW is applied on "few-shot learners".

* On line 351-352, there is "In addition, batch size warmup has a limitation that the batch size must be multiple of data-parallel size, which will be large for distributed training." May I know, if the proposed SLW has similar "limitation"? It seems like the "sequence length" also "must be multiple of data-parallel size which will be large for distributed training" if want to use the Tensor Core acceleration, like

* A similar phenomenon in the original BERT paper, "To speed up pre-traing in our experiments, we pre-train the model with sequence length of 128 for 90% of the steps. Then, we train the rest 10% of the steps of sequence of 512 to learn the positional embeddings." The increasing from sequence length 128 to 512 makes me also believe that the proposed SLW works to some degree. However, the official BERT believes that the longer sequence is to "learn the positional embeddings". I am not totally using this to contradict this submission,  just combining with the point one i mentioned in the weakness part above: good stability does not sufficiently leads to better accuracy performance. Hence, may I know why a smoother learning curve (called stability in this submission) is important? Because the main goal is to have a pre-trained model with higher accuracy and better generalization ability (even if it might have a little spike in the loss curve of pretraining).

**Limitations:**

* Given that the stability (means the smoothness of training loss curve in this submission) does not sufficiently leads to better accuracy and generalization, the importance of a stability (smoothness) needs better illustrated.

* For the GPT models, like the paper title of GPT-3 "language models are few-shot learners", reporting the accuracy performance of pre-trained language model using SLW, in the few-shot setting for the downstream tasks, would make the submission more convincing. Or at least give us a better picture of how well the proposed SLW is.

* If adding experiment results on other large language models, the submission would be more convincing about the universality part of the SLW.

* The relation between the proposed SLW and Curriculum learning are not that strong, as shorter sequence are not easier for sure.

**Strengths And Weaknesses:**

Strengths:
* simple and effective method for addressing the "stability-efficiency dilemma for GPT model pre-training"

Weaknesses:
* The stability (means the smoothness of training loss curve in this submission) typically does not sufficiently leads to better accuracy performance. Like in this paper, the SLW can "retrain 99% of the zero-shot accuracy", with smoother training loss curve.
* How about the few-shot performance of the pre-trained model using SLW? Adding the few-shot results (no matter higher or lower), would give us a more clear picture of how well SLW is. Because just like in the title of GPT-3 paper, "Language models are few-shot learners", we need to know how well SLW is in the few-shot settings, if SLW is applied on "few-shot learners".
* I agree that this work is based on good reproduction of the pre-training of GPT-2 and GPT-3, with thinking and new findings (SLW and stability-efficiency dilemma). How general is SLW might be an issue, for a claim, the name of which seems to be an optimizer contribution), some additional experiments on other large language models would make this paper more convincing.

---

> ### Author Response · Authors · 2022-07-29
> **Response part 1**
>
> Thank you for your comments and below are our replies.
>
> <Comment 1> "How about the few-shot performance of the pre-trained model using SLW? Adding the few-shot results (no matter higher or lower), would give us a more clear picture of how well SLW is. Because just like in the title of GPT-3 paper, "Language models are few-shot learners", we need to know how well SLW is in the few-shot settings, if SLW is applied on "few-shot learners"."
>
> "For the GPT models, like the paper title of GPT-3 "language models are few-shot learners", reporting the accuracy performance of pre-trained language model using SLW, in the few-shot setting for the downstream tasks, would make the submission more convincing. Or at least give us a better picture of how well the proposed SLW is."
>
> <Reply 1> After paper submission we actually did perform some studies of few-shot evaluation on a larger GPT-3 1.3B model, and we have just added it to the revised supplementary material A.5. Results show that similar to the original GPT-3, under few-shot prompts the average accuracy is better than zero-shot results for both models trained with baseline batch size warmup (from 41.6 to 44.8) and proposed SLW method (from 41.9 to 45.3).
>
> <Comment 2> "The stability (means the smoothness of training loss curve in this submission) typically does not sufficiently leads to better accuracy performance. Like in this paper, the SLW can "retain 99% of the zero-shot accuracy", with smoother training loss curve."
>
> "A similar phenomenon in the original BERT paper, "To speed up pre-traing in our experiments, we pre-train the model with sequence length of 128 for 90% of the steps. Then, we train the rest 10% of the steps of sequence of 512 to learn the positional embeddings." The increasing from sequence length 128 to 512 makes me also believe that the proposed SLW works to some degree. However, the official BERT believes that the longer sequence is to "learn the positional embeddings". I am not totally using this to contradict this submission, just combining with the point one i mentioned in the weakness part above: good stability does not sufficiently leads to better accuracy performance. Hence, may I know why a smoother learning curve (called stability in this submission) is important? Because the main goal is to have a pre-trained model with higher accuracy and better generalization ability (even if it might have a little spike in the loss curve of pretraining)."
>
> "Given that the stability (means the smoothness of training loss curve in this submission) does not sufficiently leads to better accuracy and generalization, the importance of a stability (smoothness) needs better illustrated."
>
> <Reply 2> For the "retain 99% of the zero-shot accuracy" GPT-3 125M model case, the SLW method didn't provide better accuracy because this experiment is an extreme case: only 10% of training data is allowed to maximize training data/time saving from baseline. For the GPT-2 cases where SLW provides accuracy improvement (see next paragraph), the highest data saving is 2.2x, much less than the 10x in GPT-3 case. In addition, in the GPT-3 1.3B model experiment added in the revised supplementary material A.5, under the same 300B training tokens the proposed SLW method provides better average accuracy (zero-shot from 41.6 to 41.9, few-shot from 44.8 to 45.3) than the baseline, demonstrating that the proposed method (in addition to the stability-efficiency benefit) can provide better accuracy performance.
>
> For the GPT-2 cases, results show that (1) All cases with the proposed SLW method provide better stability (Table 1) and WikiText PPL and LAMBADA accuracy (Table 2) than baseline. (2) When applying a larger batch size and LR to baseline (Table 2 case 4 vs case 1, case 13 vs case 10), the WikiText/LAMBADA performance becomes worse due to instability issues (Table 1 case 3 vs case 1, case 9 vs case 7). On the other hand, the proposed method provides better accuracy performance (Table 2 cases 5, 6, 14, 15) since it provides stable training on larger batch size and LR (Table 1 cases 4 and 10). For the BERT paper, we feel that it would be hard to conclude given that there is no evaluation of the case where 100% of steps use sequence length 512.
>
> To conclude, results demonstrate that the proposed SLW method, given a fair amount of training data (i.e., excluding cases where the data is extremely less than baseline), can provide both better stability and better accuracy performance compared to the baseline. In the final version, we will include a discussion to make this clearer.

---

> ### Author Response · Authors · 2022-07-29
> **Response part 2**
>
> <Comment 3> "I agree that this work is based on good reproduction of the pre-training of GPT-2 and GPT-3, with thinking and new findings (SLW and stability-efficiency dilemma). How general is SLW might be an issue, for a claim, the name of which seems to be an optimizer contribution), some additional experiments on other large language models would make this paper more convincing."
>
> "If adding experiment results on other large language models, the submission would be more convincing about the universality part of the SLW."
>
> <Reply 3> Our work focused on GPT-2 and GPT-3 since the GPT model family currently has the largest size, highest training cost, and severe training instability issues at large-scale (https://twitter.com/Thom_Wolf/status/1447565680384032776, https://medium.com/yandex/yandex-publishes-yalm-100b-its-the-largest-gpt-like-neural-network-in-open-source-d1df53d0e9a6). We did mention a related work that applies sequence length-based curriculum learning to BERT ("Reducing bert computation by padding removal and curriculum learning."), but it focused only on training efficiency. It would be interesting to apply our work to BERT/other models when they exhibit more instability issues as their scale increases.
>
> <Comment 4> "On line 351-352, there is "In addition, batch size warmup has a limitation that the batch size must be multiple of data-parallel size, which will be large for distributed training." May I know, if the proposed SLW has similar "limitation"? It seems like the "sequence length" also "must be multiple of data-parallel size which will be large for distributed training" if want to use the Tensor Core acceleration, like"
>
> <Reply 4> The two limitations are actually quite different: for the proposed SLW method it's a fixed "multiple of 8" limitation to leverage NVIDIA's Tensor Core. For batch size warmup it's a dynamic "multiple of data-parallel size" limitation: since nowadays pre-training tasks are performed on up to thousands of GPUs, the data-parallel size can easily go beyond 100, prohibiting flexible configuration of the method (or requires reducing the number of GPUs when using smaller batches, increasing the training clock time).
>
> <Comment 5> "The relation between the proposed SLW and Curriculum learning are not that strong, as shorter sequence are not easier for sure."
>
> <Reply 5> As mentioned in the footnote on paper page 3, "The shorter sequences are not necessarily easier but can be viewed as simpler examples since there are less context to embed.". Simpler examples are also regarded as a kind of curriculum learning (such as in the original Curriculum Learning paper, Bengio et al), and sequence length was used in curriculum learning works (but unrelated to training stability). On the other hand, we agree that the proposed SLW method is different from traditional curriculum learning as discussed in Section 2. In the final version, we will clarify this relationship more clearly.

---

### Official Review · Reviewer_e6nP · 2022-07-18

**Rating:** 8
**Confidence:** 3
**Soundness:** 3 good
**Presentation:** 3 good
**Contribution:** 3 good

**Summary:**

This paper names and addresses the stability/efficiency dilemma in large language model training: larger batch sizes and learning rates enable more efficient and faster distributed training, but also increase the prevalence of instabilities which tend to hurt performance and can even cause divergence (Figures 5 and 6).  This problem appears to get worse for larger models.

The work proposes increasing the sequence length throughout the initial stage of training (“Sequence Length Warmup”).  This was proposed already in ShortFormer, but here they use a more sophisticated schedule, describe how to choose the schedule, and provide an explanation and analysis of this method in terms of instability.

Their experiments involve training open-source versions of GPT-2 and (125M) GPT-3, and produce large efficiency gains.

**Questions:**

Questions:
* Do you in fact tune the duration hyperparameter on the test set? (251-252)  I assume it is the validation set?  Else this is a major problem for the soundness of the work.
* Would the approach mentioned on lines 230-234 scale to data limited regimes?  If I understand correctly, you are throwing away data when you truncate sequences?
* RE 263-265: If I understand correctly, it was necessary to train to convergence in order to discover this heuristic? Did you verify that this heuristic works in other cases in order to validate it?
* Why do you say the method acts as a regularizer? (line 330)  This claim seems unsupported and is never discussed elsewhere that I noticed.

Suggestions:
* “gradient variance” should be defined/explained more thoroughly.
* The document should be checked for grammar and typos (e.g. 32, 64, 84).
* A few cite’s should be citet’s (e.g. [37] in Table 2).
* Saying the methods “solves” the dilemma (217) is overstated; e.g. it should be qualified with “in our experiments” or something like that.

**Limitations:**

Large language models are liable to have a large impact on society; this impact may not be positive.  Increasing the efficiency of training LLMs is likely to increase deployment, increase the size of models deployed, and facilitate more study of the impact and potential harms of these models.  This could be worth discussing briefly.

**Strengths And Weaknesses:**

Strengths:
* The proposed method appears to work very well and produce major gains in efficiency.
* There are strong supporting results demonstrating that the method removes instabilities.
* There is a good and clear discussion of the experiments and method.  Overall clarity was good.

Weaknesses:
* The reason why using longer sequences creates instability was not discussed much.
* The reason why instabilities lead to worse performance when they do not cause divergence was not discussed much.
* The existence of ShortFormer detracts a bit from the novelty.

---

> ### Author Response · Authors · 2022-07-29
> **Response**
>
> Thank you for your comments and below are our replies.
>
> <Comment 1> "The reason why using longer sequences creates instability was not discussed much."
>
> <Reply 1> We agree that although our paper discovers the correlation between long sequences and training instability (and proposes a method to alleviate the issue), the root cause of this causal relationship is not fully deciphered and would be an interesting future work direction. One assumption we have, as mentioned in the footnote on page 3, is that "The shorter sequences are not necessarily easier but can be viewed as simpler examples since there are less context to embed.". We think encoding shorter sequences (especially at the early training stage when weight is still relatively random) will generate less "noisy" gradients thus leading to higher stability. In the final version, we will add a better summary of our conjectures.
>
> <Comment 2> "The reason why instabilities lead to worse performance when they do not cause divergence was not discussed much."
>
> <Reply 2> We agree that a discussion is necessary and will add one in the final version. In short, our experiments show that training instabilities lead to loss spikes, and it requires a nontrivial number of training steps/tokens to recover the training loss back to a normal level (e.g., Figure 1(b)). These training "detours" slow down the learning and eventually lead to worse accuracy performance (e.g., Table 2 case 13 vs case 10).
>
> <Comment 3> "The existence of ShortFormer detracts a bit from the novelty."
>
> <Reply 3> We agree that ShortFormer is a closely related work, but we believe that our work demonstrates novel contributions as described in paper lines 54-64.
>
> <Comment 4> "Do you in fact tune the duration hyperparameter on the test set? (251-252) I assume it is the validation set? Else this is a major problem for the soundness of the work."
>
> <Reply 4> We agree the description is confusing and will fix it in the final version. First, our tuning strategy (used for GPT-2 1.3B and GPT-3 125M cases) relies only on the validation set. On the other hand, to design and verify this tuning strategy, for GPT-2 117M case we performed multiple full training with different duration hyperparameters and performed test set/WikiText/LAMBADA evaluations after full training to confirm which duration provides the best accuracy performance.
>
> <Comment 5> "Would the approach mentioned on lines 230-234 scale to data limited regimes? If I understand correctly, you are throwing away data when you truncate sequences?"
>
> <Reply 5> It is true that the truncation implementation will drop some data in the current step. However, with some implementation changes, it's possible to memorize the dropped data and use them in future steps. In addition, we are also exploring different implementations of the proposed SLW method. We will add this discussion in the final version.
>
> <Comment 6> "RE 263-265: If I understand correctly, it was necessary to train to convergence in order to discover this heuristic? Did you verify that this heuristic works in other cases in order to validate it?"
>
> <Reply 6> Yes for GPT-2 117M case we train to convergence to discover the tuning heuristic. We then apply this tuning strategy to GPT-2 1.3B and GPT-3 125M cases and observed similar training efficiency benefits compared to the GPT-2 117M case. Due to time/hardware resource limitations, we didn't perform the full training grid search on GPT-2 1.3B and GPT-3 125M cases.
>
> <Comment 7> "Why do you say the method acts as a regularizer? (line 330) This claim seems unsupported and is never discussed elsewhere that I noticed."
>
> <Reply 7> We agree that this part is more like a preliminary conjecture and is a bit off-topic from the main paper. We will remove it in the final version.
>
> <Comment 8> "“gradient variance” should be defined/explained more thoroughly."
>
> "The document should be checked for grammar and typos (e.g. 32, 64, 84)."
>
> "A few cite’s should be citet’s (e.g. [37] in Table 2)."
>
> "Saying the methods “solves” the dilemma (217) is overstated; e.g. it should be qualified with “in our experiments” or something like that."
>
> "Large language models are liable to have a large impact on society; this impact may not be positive. Increasing the efficiency of training LLMs is likely to increase deployment, increase the size of models deployed, and facilitate more study of the impact and potential harms of these models. This could be worth discussing briefly."
>
> <Reply 8> We agree and will apply all of these suggestions in the final version.

---

### Meta-Review · Area_Chair_mjFh · 2022-08-28

**Recommendation:** Accept
**Confidence:** Certain

**Metareview:**

# Summary of the Paper

This paper makes two contributions as far as I can tell:
1. The paper attempts to characterize training instability in large language models.
2. The paper presents Sequence Length Warmup (SLW), a technique that the authors claim to reduce training instability and that empirically makes training LLMs much more efficient.

# Metareview

The paper isn't perfect, but it's an obvious accept. Even in the most pessimistic assessment of the paper, the SLW technique is a clear win for efficient training regardless of how harshly you judge the analysis of training instability.

The biggest weaknesses of the paper are (1) all of the methodological and scientific questions around trying to grapple with the phenomena around training stability and (2) claims of a connection between SLW and training instability.

On the topic of instability in large model training (1), there has been plenty of griping on Twitter but very little scientific analysis of the phenomenon. This paper's analysis will hardly be the last word on that topic, but it's a reasonable first attempt at something that other researchers will doubtlessly build on and hone over the coming years. I share Reviewer c8jb's concerns that correlation with gradient norms is insufficient to make claims about training stability or SLW and that more rigorous definitions of training stability are necessary. However, even if, in the very worst-case scenario, this paper is one of the first attempts at characterizing a vexing phenomenon and inspires future researchers to tear apart the modes of analysis and findings in this paper to improve upon them, this paper will have been a worthwhile contribution to the scholarly literature.

On the topic of claims of a connection between SLW and training instability (with the concerns most poignantly expressed by Reviewer S5Qt), this is really a byproduct of the scientific analysis of training instability. This connection (or lack thereof) will become clearer as time goes on and the science improves, and I simply ask that the authors acknowledge the uncertainty here.

**If the above scenarios are the very worst possible outcome for this paper, it's still a major contribution to both science and practice. I therefore advocate for accepting this paper, and the reviewers seem to agree with that assessment (both the good and the bad). I urge the authors to prepare the camera-ready version of the paper by carefully incorporating the feedback of the reviewers and, in particular, Reviewer c8jb, who had some very thoughtful comments about ways to clarify the scientific aspects of the paper. To satisfy the reviewers and future readers, I highly recommend openly and frequently acknowledging the vast uncertainty we have about the scientific aspects of this paper as we strive to make sense of this strange training instability phenomenon.**

The reviewers were enthusiastic and engaged, and the discussion was lively, which suggests to me that this is an exciting paper that deserves to be featured to the community via publication at NeurIPS.

There were some other common comments that I urge the authors to address in order to produce the best and most influential possible version of this paper:
* "The reason why instabilities lead to worse performance when they do not cause divergence was not discussed much.", "The reason why using longer sequences creates instability was not discussed much." The authors discussed this a bit during the discussion period, but it's worth emphasizing these as open questions so that other researchers know it's important to follow up on.
* It's worth being crystal clear about truncation vs. just focusing on shorter sequences. It's an important experimental detail that may seem surprising or counterintuitive if it's not clearly laid out.

**Award:**

No

---

### Decision · Program_Chairs · 2022-09-14

Accept